# Cardiolipin inhibits the non-canonical inflammasome by preventing LPS binding to caspase-4/11

Malvina Pizzuto [ID] [1,2,3 ✉], Mercedes Monteleone [ID] [1,9], Sabrina Sofia Burgener [ID] [1,9], Jakub Begas [4], Melan Kurera [5], Jing Rong Chia [1], Emmanuelle Frampton [1], Joanna Crawford [ID] [1], Monalisa Oliveira [ID] [1], Kirsten M Kenney [1], Jared R Coombs [ID] [1], Masahiro Yamamoto [ID] [6,7], Si Ming Man [ID] [5], Petr Broz [ID] [4], Pablo Pelegrin [ID] [2,8,10] & Kate Schroder [ID] [1,10]

## Abstract

**Caspase-4 and caspase-11 (CASP4/11) sense bacterial lipopolysaccharide (LPS). Currently available inhibitors of CASP4/11 also block the activity of caspase-1 (CASP1), which restricts their usefulness in the study of CASP4/11 functions, as well as their clinical potential for the treatment of LPS-linked diseases through CASP4/11 inhibition. Here, we identify mitochondrial cardiolipin as a selective inhibitor of CASP4/11-dependent cell death and inflammatory cytokine secretion, without affecting CASP1 function. Cardiolipin targets the CARD domain of CASP4/11, impeding its interaction with LPS to restrain CASP4/11 activation, thereby suppressing LPS-induced systemic inflammation in vivo. By identifying cardiolipin as a selective inhibitor of CASP4/11, we provide an urgently needed tool for studying caspase-4/11 and non-canonical inflammasome functions in inflammatory pathways and LPS-induced pathogenesis.**

**Keywords** Cardiolipin; Noncanonical Inflammasome; Caspase-4; Caspase-11; LPS
**Subject Categories** Autophagy & Cell Death; Immunology

## Introduction

The noncanonical inflammasome is a pivotal component of the innate immune system, responsible for sensing intracellular bacterial lipopolysaccharides (LPS) through caspase-4 in humans and caspase-11 in mice (hereafter CASP4/11). Unlike the canonical inflammasome, which relies on sensor proteins to activate the effector protein caspase-1 (CASP1) (Schroder and Tschopp, 2010), in the noncanonical inflammasome, CASP4/11 are directly activated by LPS, with these caspases acting as both sensors and effector proteins (Kayagaki et al, 2011; Schmid-Burgk et al, 2015). Upon binding to LPS, CASP4/11 homodimerise and auto-cleave to form their active protease species, which cleave the pore-forming protein gasdermin D (GSDMD) (Chan et al, 2023; Kajiwara et al, 2014; Lee et al, 2018; Ross et al, 2018). CASP4, but not CASP11, also cleaves the inactive pro-forms of interleukin (IL)-1β and IL-18 to allow the mature cytokines to exit the cell through the GSDMD pores (Chan et al, 2023; X Shi et al, 2023). Eventually, GSDMD pores induce a lytic form of cell death called pyroptosis (Chan et al, 2023; Kajiwara et al, 2014; Lee et al, 2018; Ross et al, 2018). Pyroptosis ejects intracellular contents such as lactate dehydrogenase (LDH) and soluble pro-inflammatory damage-associated molecular patterns (DAMPs) into the extracellular space (Broz et al, 2019) while leaving behind a corpse that is immunogenic to dendritic cells (Holley et al, 2025).

Extracellular LPS binds to its transmembrane receptor, Toll-like Receptor 4 (TLR4), which leads to the production and release of inflammatory cytokines, including tumour necrosis factor (TNF), IL-6, and interferon (IFN)-β (Franz and Kagan, 2017; Park et al, 2009). TLR4 activation also induces the expression of proteins involved in the canonical and noncanonical inflammasome pathways, including the inflammasome sensors nucleotide-binding domain leucine-rich repeat and pyrin domain-containing protein 3 (NLRP3), absent in melanoma 2 (AIM2), and CASP11, as well as the inflammasome substrate pro-IL-1β (Schroder and Tschopp, 2010). This poises cells to respond to microbial threats by activating inflammasomes to produce cytokines (e.g., IL-1β, IL-18) essential for antimicrobial defence (Schroder and Tschopp, 2010). In the noncanonical inflammasome, CASP4/11-induced GSDMD pores trigger potassium efflux that activates the NLRP3 inflammasome, causing CASP1 activation and further IL-1β, IL-18, and GSDMD processing that amplify the inflammatory response (Baker et al, 2015; Kayagaki et al, 2015; Schmid-Burgk et al, 2015; Viganò et al, 2015).

[1]Institute for Molecular Bioscience, The University of Queensland, Brisbane, QLD 4072, Australia. [2]Molecular Inflammation Group, Biomedical Research Institute of Murcia (IMIB-Arrixaca), Murcia, Spain. [3]Structure and Function of Biological Membranes Laboratory, Université Libre de Bruxelles, Brussels, Belgium. [4]Department of Immunobiology, University of Lausanne, Epalinges, Switzerland. [5]Division of Immunology and Infectious Diseases, The John Curtin School of Medical Research, The Australian National University, Canberra, ACT, Australia. [6]Department of Immunoparasitology, Research Institute for Microbial Diseases Osaka University, Suita, Osaka, Japan. [7]Laboratory of Immunoparasitology, WPI Immunology Frontier Research Center Osaka University, Suita, Osaka, Japan. [8]Department of Biochemistry and Molecular Biology B and Immunology, Faculty of Medicine, University of Murcia, Murcia, Spain. [9]These authors contributed equally: Mercedes Monteleone, Sabrina Sofia Burgener. [10]These authors contributed equally: Pablo Pelegrin, Kate Schroder. ✉E-mail: m.pizzuto@uq.edu.au

CASP4/11 activation by LPS in macrophages and epithelial cells is vital to detect bacterial threats and mount an immune response (Aachoui et al, 2013, 2015; KW Chen et al, 2018; Enosi Tuipulotu et al, 2023; Knodler et al, 2014; Kobayashi et al, 2013; Kovacs et al, 2020; Kumari et al, 2021; Wang et al, 2018). However, aberrant CASP4/11 activation may be detrimental (R Chen et al, 2019; Cheng et al, 2017; Deng et al, 2018; Hagar et al, 2013; Kajiwara et al, 2014; Kayagaki et al, 2011, 2015; Tang et al, 2018; Wei et al, 2024). In preclinical models of sepsis, a life-threatening condition caused by a dysregulated host response to infection (Singer et al, 2016), signal blockade of CASP4 or CASP11 protected from organ damage and death (Cheng et al, 2017; Deng et al, 2018; Hagar et al, 2013; Kayagaki et al, 2013; Wei et al, 2024). However, clinical trials that blocked IL-1β signalling were unsuccessful (Dinarello, 2018; Marshall, 2014), perhaps because these also blocked CASP1-mediated antimicrobial defence. Indeed, immunosuppressed patients who cannot mount a canonical NLRP3/CASP1 inflammasome response have the highest sepsis mortality rates (Martínez-García et al, 2019). While such reports highlight the potential benefit of specific inhibition of CASP4/11 for treating sepsis, such CASP4/11-specific inhibitors remain to be identified.

Due to the dearth of specific CASP4/11 inhibitors, genetic deletion remains the primary experimental approach for investigating CASP4/11 functions. By contrast, NLRP3 function is easily dissected using a wide range of inhibitors (Coll et al, 2022). Thus, despite its intricate interconnection with the NLRP3/CASP1 inflammasome, the role of CASP4/11 in inflammatory pathways and diseases remains challenging to investigate, with clinical trials predominantly focused on targeting NLRP3 inhibition (Coll et al, 2022).

CASP1 and CASP4/11 are enzymes that contain a caspase-recruitment and activation domain (CARD) followed by a protease domain (PD) (Chan et al, 2023; Ross et al, 2018). The CARD mediates signal sensing and caspase dimerisation, while the PD is responsible for caspase autocleavage, as well as binding and cleavage of the substrates (Chan et al, 2023; Ross et al, 2018). Inhibitors that block the CASP4/11 PD also inhibit CASP1 due to similarities within the PDs of these caspases (Ekert et al, 1999; Green, 2022). By contrast, the CASP4/11 CARD domain differs substantially from CASP1, as the CASP4/11 CARD binds LPS while the CASP1 CARD does not (Devant et al, 2021; J Shi et al, 2014). Thus, preventing LPS binding to the CASP4/11 CARD may be a useful strategy to develop CASP4/11-specific inhibitors. Research by us and others showed that di-unsaturated cardiolipin with an 18-carbon atom chain length (hereafter called CL) inhibits TLR4 (Balasubramanian et al, 2015; Pizzuto et al, 2019; Wenzel et al, 2021). We showed that unsaturated CLs specifically inhibit TLR4 signalling without affecting signalling by other TLRs, while saturated CLs activate TLR4 (Pizzuto et al, 2019). Competition tests and molecular docking suggested that CL likely occupies the LPS-binding site of TLR4 (Pizzuto et al, 2019). If binding studies confirm this mechanism-of-action, it would rule out other possible activities, such as promoting CD14 endocytosis (Tan et al, 2015). The TLR4 antagonists, LPS derived from *R. sphaeroides* (RS-LPS) and its analogue eritoran, are lipids that block the LPS-binding site on TLR4 and possess a single mono-unsaturated chain, and are thus similar to CL.

Here, we examined whether these competitive TLR4 antagonists might also compete with LPS for binding CASP4/11, thereby acting as specific inhibitors of CASP4/11 over CASP1. To test this, we examined the ability of natural unsaturated CL, saturated CLs, and RS-LPS to inhibit CASP4/11 activation. We found that di-unsaturated CL, but not RS-LPS or saturated CLs, is a CASP4/11 inhibitor. Mechanistically, CL targets the CASP4/11 CARD domain, preventing LPS binding and resulting CASP4/11 signalling. CL specifically inhibits CASP4/11 over CASP1, dampening cell death and inflammatory cytokine secretion in vitro and in vivo. These findings identify CL as a novel and specific CASP4/11 inhibitor for studying the physiological and pathological functions of CASP4/11 in immune pathways. We showed that di-unsaturated CL dampens cell death and cytokine secretion, while mono-unsaturated LPS or saturated CLs do not. This study gives new molecular insights into caspase regulation by lipids, offering new avenues for translation to urgently needed novel treatments for human sepsis.

## Results and discussion

### Unsaturated cardiolipin, but not saturated cardiolipins or RS-LPS, inhibits caspase-4/11 signalling

We first sought to determine whether di-unsaturated 18:2 CL, saturated 16:0 or 18:0 CLs, or mono-unsaturated RS-LPS suppressed CASP4/11 noncanonical inflammasome signalling outputs. We treated Pam₃CSK₄ (Pam)-primed human monocyte-derived macrophages (HMDM) or bone marrow-derived murine macrophages (BMDM) with the noncanonical inflammasome activator LPS from *E. coli* B4, delivered intracellularly using lipofectamine (LTX) or Cholera toxin B (CTB), in the absence or presence of CLs or RS-LPS. We quantified IL-1β and LDH release as a measure of noncanonical inflammasome signalling induced by intracellular LPS (iLPS). iLPS induced IL-1β and LDH release in HMDM and BMDM, and this was unaffected by saturated CL (Fig. EV1), suppressed by unsaturated CL, and potentiated by RS-LPS (Fig. 1A,B). This indicates that unsaturated CL dampens noncanonical inflammasome signalling and suggests that possession of saturated chains or only one mono-unsaturated chain is insufficient for tetra-acylated lipids (e.g., CL, RS-LPS, and its analogue eritoran) to inhibit CASP4/11, although RS-LPS and saturated di-acylated lipids can bind CASP4/11 (Cao et al, 2025; J Shi et al, 2014). The findings that RS-LPS boosted iLPS-induced CASP4/11 signalling, and RS-LPS alone triggered LDH and IL-1β release in BMDM, are in line with a report that RS-LPS induced LDH and IL-1β release in macrophages (Lagrange et al, 2018) and in contrast with others that showed RS-LPS did not induce CASP4/11 activation or oligomerisation (Cao et al, 2025; J Shi et al, 2014). To investigate how RS-LPS affected TLR4-induced NF-κB activation, we treated BMDM and HMDM with RS-LPS without other stimuli for 4 h and measured TNF secretion. RS-LPS induced TNF release in BMDM but not in HMDM (Appendix Fig. S1), suggesting that BMDM exposure to RS-LPS in the absence of other stimuli may trigger NF-κB activation. This was unexpected because RS-LPS is described as a TLR4 antagonist (Rose et al, 1995). To further investigate this, we treated macrophages with RS-LPS in the presence or absence of LPS. In both BMDM and HMDM, RS-LPS inhibited TNF release induced by extracellular LPS (Appendix Fig. S1), confirming the quality of our ultrapure RS-LPS preparation and its reported function as an antagonist of LPS-TLR4 signalling. These data collectively suggest that while RS-LPS is an antagonist of human TLR4, RS-LPS could be a partial agonist for murine TLR4. This is not unprecedented amongst lipid

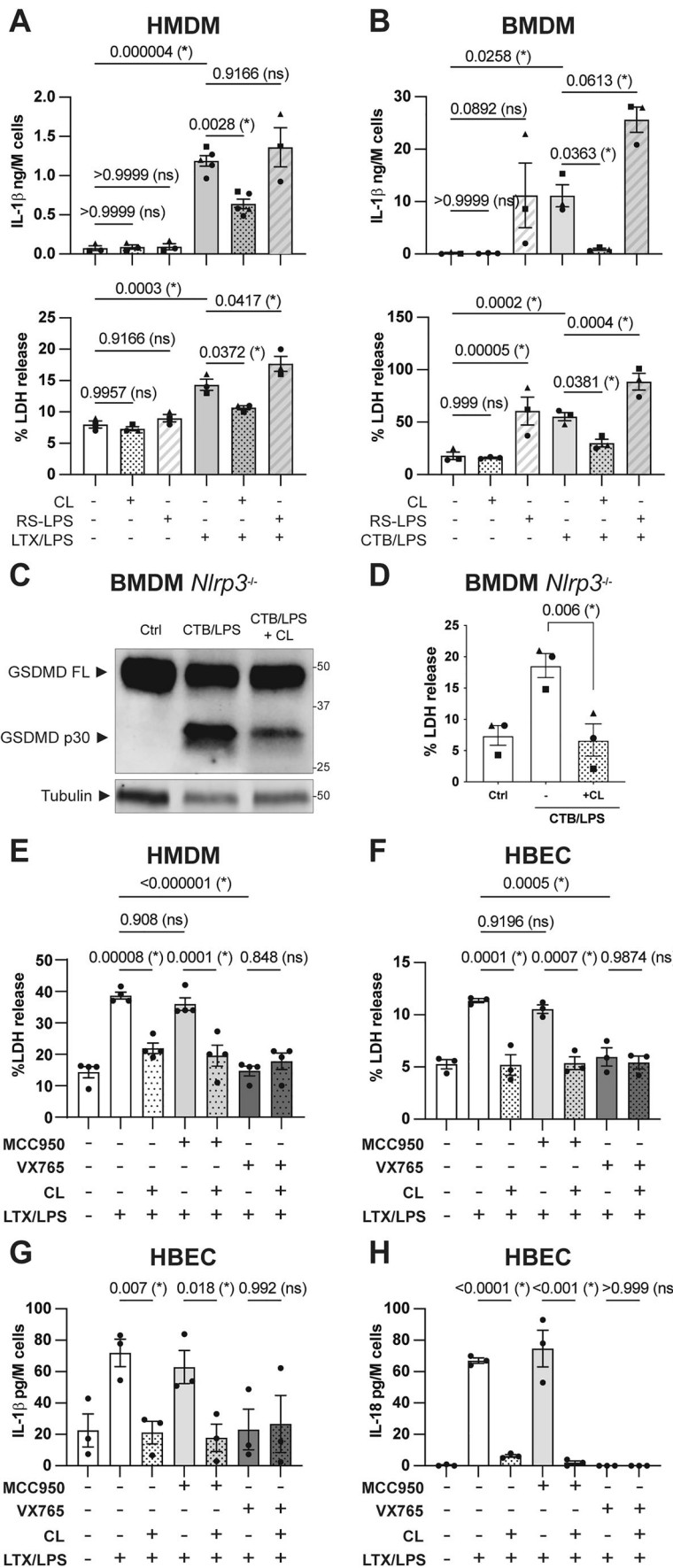

**Figure 1.** CL but not RS-LPS inhibits noncanonical inflammasome signalling independently of NLRP3 in primed macrophages and epithelial cells.

(A) Human monocyte-derived macrophages (HMDM) from healthy donors or (B) Bone marrow-derived macrophages (BMDM) from wild-type (WT) mice were incubated for 4 h with 1 µg/mL $Pam_3CSK_4$. Cell culture medium was replaced with OptiMEM plus HEPES (−), 10 µM CL, or 5 µg/mL of RS-LPS (complexed with LTX) in the absence or presence of 2 µg/mL of LPS complexed with LTX (A) or CTB (B). Cells were incubated for 4 h (A) or 18 h (B). Cell supernatants were analysed for cleaved IL-1β (ELISA) and LDH (cytotoxicity assay). (C, D) BMDM from $Nlrp3^{−/−}$ mice were incubated for 4 h with 1 µg/mL $Pam_3CSK_4$ before cell culture medium was replaced with OptiMEM plus HEPES (Ctrl), or 2 µg/mL of LPS complexed with CTB in the presence of HEPES (−), or 10 µM CL (CL). Cells were incubated for 18 h. GSDMD cleavage and tubulin expression were assessed in mixed supernatants and lysates by western blot (C). LDH release was quantified by cytotoxicity assay (D). (E) HMDM from healthy donors or (F–H) Human bronchial epithelial cells (HBEC) were incubated for 4 h (E) or 18 h (F–H) with 1 µg/mL $Pam_3CSK_4$. Cell culture medium was replaced with OptiMEM with vehicle (DMSO) or 10 µM MCC950 (NLRP3 inhibitor), or 10 µM VX765 (CASP1/4 inhibitor). Cells were incubated for 1 h, and then 2 µg/mL LPS complexed with LTX was added in the presence of HEPES (−) or 10 µM CL (CL). Cells were incubated for 4 h. Cell supernatants were analysed for LDH (E, F) (cytotoxicity assay), cleaved IL-1β (G) and IL-18 (H) (ELISA). Data information: Each symbol is the mean of technical triplicates from an independent biological replicate. Bars are the mean of three or more independent biological replicates ($n = 3$–5) ± SEM. Statistical analysis: Data were verified for normality using a Shapiro–Wilk test and analysed by (A, B, E–H) one-way ANOVA Šídák's multiple comparisons test, (D) paired $t$ test. $P$ values are reported above bars. Statistical significance was defined as follows: significant difference for $P < 0.05$ (*), not significant for $P \geq 0.05$ (ns). Source data are available online for this figure.

modulators of TLR4; for example, lipid IVa is also a partial agonist of murine TLR4 but an antagonist of human TLR4 (Walsh et al, 2008). Given that NF-κB signalling licenses CASP11 for signalling, RS-LPS-induced NF-κB activity provides a potential explanation for how RS-LPS boosts iLPS-induced CASP11 signalling in BMDM. Whether RS-LPS also activates the noncanonical inflammasome remains an open question for future studies.

CL was previously reported to induce CASP1 cleavage in lysates from a macrophage cell line (Iyer et al, 2013). To ensure that inhibition of iLPS signalling was not due to CL-induced CASP1-dependent cell death, we treated primed HMDM and BMDM with CL alone. In both primed HMDM and BMDM, CL failed to induce the release of IL-1β or LDH (Fig. 1A,B), suggesting that CL does not activate inflammasome signalling in primary human and murine macrophages.

We also tested that CL was not toxic to macrophages by treating primed BMDM or HMDM with escalating doses of CL (from 1 to 50 µM). Cell toxicity was monitored by measuring LDH release and YoPro uptake, all of which confirmed that CL did not induce macrophage cell death (Fig. EV2). Instead, increasing CL doses further decreased iLPS-induced LDH and IL-1β release, resulting in an $IC_{50}$ of around 5 µM (Appendix Fig. S2).

Then, to determine whether CL inhibits noncanonical inflammasome signalling to a broad repertoire of LPS species, we treated Pam-primed BMDM with intracellular LPS derived from several bacterial strains (*E. coli* K12, *Salmonella minnesota* R595, *Pseudomonas aeruginosa*) as compared to the *E. coli* B4 LPS previously used. CL suppressed LDH and IL-1β release induced by all LPS species (Appendix Fig. S3), indicating that CL may inhibit noncanonical inflammasome signalling to a wide variety of bacterial strains.

## Cardiolipin inhibits the noncanonical inflammasome upstream of NLRP3 activation

Given that CASP4/11 induces GSDMD cleavage and resultant NLRP3 inflammasome signalling, we next investigated whether NLRP3 is a potential target of CL. We blocked NLRP3 signalling in primed BMDM (through *Nlrp3* knockout) or HMDM (by pre-treating cells with the NLRP3 inhibitor MCC950) before treating cells with iLPS in the absence or presence of CL. CL inhibited iLPS-induced GSDMD cleavage and LDH release in $Nlrp3^{−/−}$ BMDM (Fig. 1C,D) and MCC950-treated HMDM (Fig. 1E). The CASP1/4 inhibitor VX765 blocked iLPS-induced LDH release (Fig. 1E),

confirming that HMDM death was pyroptotic. Collectively, these data demonstrate that CL inhibits noncanonical signalling in human and murine macrophages upstream and independently of NLRP3.

In murine macrophages, NLRP3/CASP1 signalling is required for IL-1β and IL-18 cleavage and release downstream of CASP11 activation (Kayagaki et al, 2015). In human cells, however, CASP4 can cleave pro-IL-1β and pro-IL-18 independently of NLRP3; accordingly, in human epithelial cells that do not express NLRP3, CASP4 activity is sufficient for the release of mature IL-1β and IL-18 (Chan et al, 2023; X Shi et al, 2023). To determine whether CL blocks CASP4-mediated pro-IL-1β and pro-IL-18 cleavage independently of NLRP3, we thus tested human bronchial epithelial cells (HBEC). CL suppressed iLPS-induced LDH, IL-1β, and IL-18 release from HBEC, and this was unaffected by MCC950 as expected (Fig. 1F–H). Verifying that these signalling outputs are inflammasome-dependent, VX765 blocked iLPS-induced LDH, IL-1β, and IL-18 release (Fig. 1F–H). Thus, CL blocks signalling by CASP4 in the absence of NLRP3. Moreover, these data extend the CASP4 inhibitory activity of CL to non-myeloid cells.

## CL binds to the CARD domain of CASP4/11, preventing LPS binding and consequent CASP4/11 activation

To define the mechanism by which CL suppresses noncanonical inflammasome signalling, we initially examined whether CL blocks CASP4/11 intracellular signalling in HMDM and BMDM. Macrophages were left unprimed (UP) or were primed with $Pam_3CSK_4$ and then treated with iLPS in the presence or absence of CL. The cell culture medium was precipitated and resuspended in cell lysates for immunoblot analyses of CASP4/11, GSDMD, IL-1β, and CASP1 cleavage. HMDM control samples (unprimed and primed) express CASP4, CASP1, and GSDMD, and their expression is not affected by priming, while pro-IL-1β was strongly induced by priming (Fig. 2A). Notably, CL did not affect the expression of these full-length proteins but suppressed iLPS-induced GSDMD, CASP1, and IL-1β cleavage (Fig. 2A). Although we were unable to detect the cleaved fragment of CASP4, iLPS induced a loss of full-length CASP4 protein that was suggestive of cleavage, and this was restored by co-incubation with CL (Fig. 2A; quantification in Appendix Fig. S4). Similarly, in BMDM, iLPS induced CASP11, IL-1β, GSDMD, and CASP1 cleavage, and this was reduced by co-incubation with CL without affecting the expression of full-length CASP11, IL-1β, GSDMD and CASP1 (Fig. 2B; quantification in

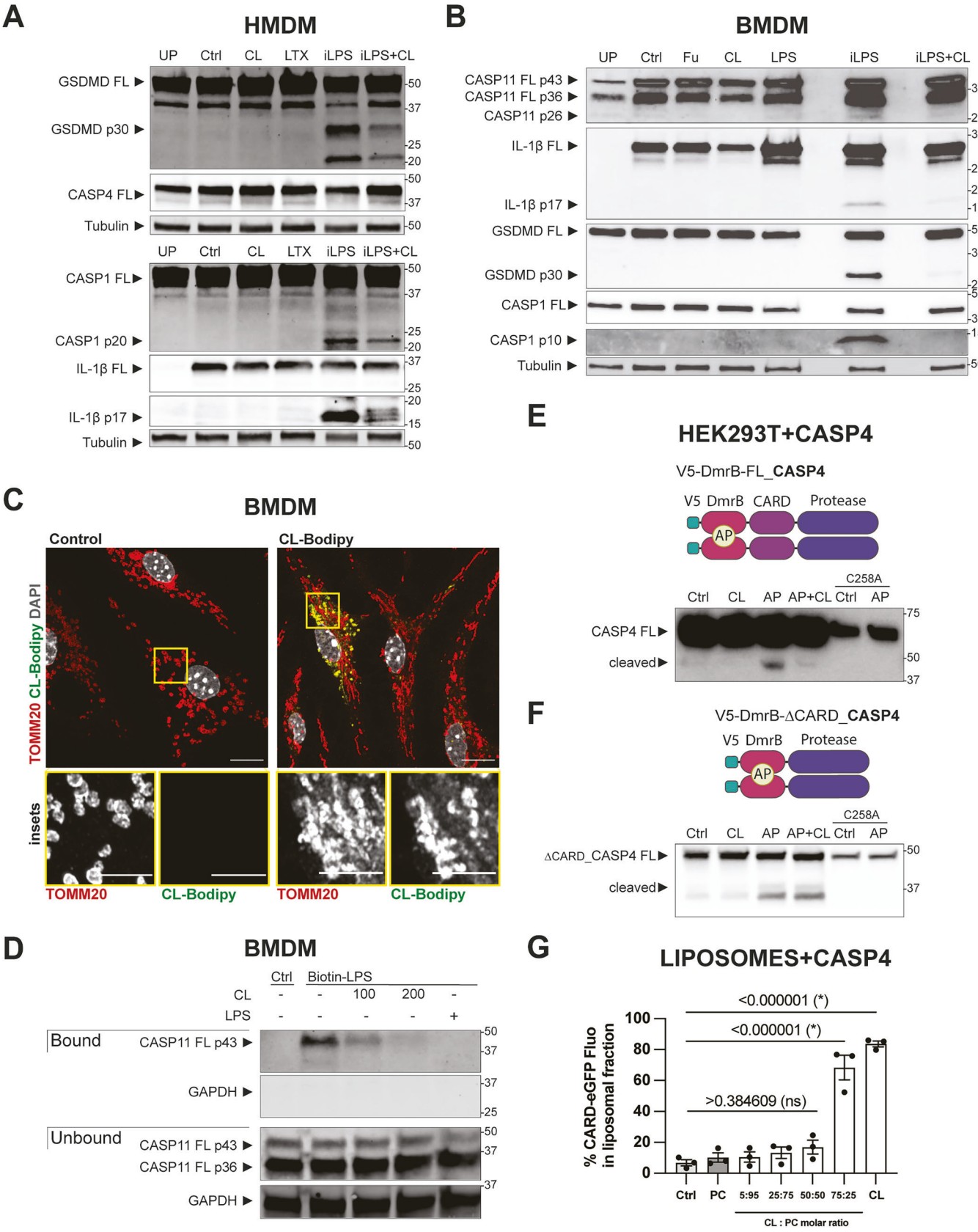

◀ **Figure 2. CL reaches the cell interior and binds to the CARD domain of CASP4/11, preventing LPS binding and consequent CASP4/11 activation.**

(A) HMDM or (B) WT BMDM were incubated for 4 h with cell culture medium (unprimed, UP) or 1 μg/mL Pam₃CSK₄ (all other conditions). Cell culture medium was then replaced with OptiMEM (Ctrl), FuGENE HD (Fu) 0.5% v/v, LTX 0.25% v/v, 10 μM CL, or 2 μg/mL of LPS complexed with 0.25% v/v LTX (A) or 0.5% FuGENE HD (B) in the presence of HEPES vehicle (iLPS) or 10 μM CL (iLPS + CL). Cells were incubated for 4 h. GSDMD, CASP1, CASP4, CASP11, IL-1β, and tubulin expression and cleavage were assessed in mixed supernatants and lysates by western blot. (C) Fixed-Airyscan confocal imaging of WT BMDM primed for 4 h with 1 μg/mL Pam₃CSK₄ and incubated with HEPES (Control) or 10 μM of CL liposomes containing 1% (w/w) TopFluor CL (BODIPY-CL) for 18 h. Macrophages were immunostained with TOMM20 (red), BODIPY-CL (green), and DAPI (grey). Images are maximum intensity projections of Z-stack acquisitions. (D) WT BMDM were incubated for 4 h with 1 μg/mL Pam₃CSK₄. Cells were then lysed and incubated with HEPES (−), 100 or 200 μg/mL CL or 100 μg/mL LPS for 1 h. 1 μg/mL of biotinylated LPS (biotin-LPS), was added, and lysates were incubated for 2 h. Biotinylated LPS with bound proteins was purified using magnetic streptavidin beads. Both purified (bound) and unbound fractions were immunoblotted for CASP11 and GAPDH by western blot. (E, F) Upper panels: Schematic of full-length (E) and ΔCARD (F) CASP4 constructs with the DmrB dimerisation system, permitting controlled dimerisation by AP20187 (AP). All DmrB constructs were N-terminally V5-tagged. (E, F) Lower panels: HEK293T cells were transfected with the native or the CASP4 catalytic cysteine mutant (C258A) of V5-DmrB-CASP4 (full length, (E)) or V5-DmrB-ΔCARD-CASP4 (F) constructs. 24 h post-transfection, cells were harvested, plated, and 4 h later treated with AP for 50 min in the absence or presence of 30 μM CL (1 h pre-incubation). CASP4 auto-processing was analysed by western blot of cell lysates. (G) Recombinant CASP4-CARD-eGFP (1 μM) was incubated for 1 h at 37 °C with HEPES (Ctrl), or 1.67 mM of liposomes containing phosphatidylcholine (PC) or CL alone (CL) or increasing CL:PC molar ratio (5:95, 25:75, 50:50 and 75:25). Samples were centrifuged and the eGFP fluorescence was measured in supernatant (unbound CARD in non-liposome fraction) and resuspended pellet (liposome-bound CARD). The Y axis represents the percentage of overall fluorescence in the pellet. Data information: Each symbol is the mean of technical triplicates from an independent biological replicate. Bars are the mean of three independent biological replicates ($n = 3$) ± SEM. Blots and images are representative of three or four independent experiments ($n = 3$–4). Scale bar = 10 μm and 5 μm in inset panels. Statistical analysis: Data were verified for normality using a Shapiro–Wilk test and analysed by one-way ANOVA, Dunnett's multiple comparisons test. P values are reported above bars. Statistical significance was defined as follows: significant difference for $P < 0.05$ (*), not significant for $P ≥ 0.05$ (ns). Source data are available online for this figure.

Appendix Fig. S5). In all, these data indicate that in human and murine macrophages, CL inhibits iLPS-induced CASP4/11 activation and resultant GSDMD, CASP1, and IL-1β cleavage.

Pam-priming did not affect CASP4 expression in HMDM but upregulated CASP11 in BMDM and pro-IL-1β in both HMDM and BMDM (Fig. 2A,B; quantification in Appendix Figs. S4 and S5), confirming previous findings for TLR-dependent regulation at the mRNA level (Schroder et al, 2012). In BMDM, pro-IL-1β expression was further upregulated by LPS, likely by TLR4 activation (Fig. 2B). To confirm that CL did not suppress CASP4/11 signalling via an indirect mechanism involving TLR4, we measured TNF secretion (Fig. EV3A,B) alongside inflammasome signalling outputs (Fig. 2A,B). CL did not affect LPS-induced TNF secretion in HMDM and BMDM (Fig. EV3A,B), in line with our previous report that CL does not inhibit TLR4 signalling induced by the LPS dose used here to activate CASP4/11 (Pizzuto et al, 2019). Further, in Pam-primed TLR4-deficient BMDM ($Tlr4^{-/-}$), CL suppressed iLPS-induced signalling outputs, including cleavage of CASP11, GSDMD, CASP1, and IL-1β (Fig. EV3C) as well as LDH and IL-1β release (Fig. EV3D,E). Thus, CL inhibits CASP11 signalling independently of TLR4.

The efficacy of LPS-induced CASP4/11 activation depends on the amount of LPS delivered into the cytosol (Hagar et al, 2013; Kayagaki et al, 2013). To address the possibility that CL interfered with LPS delivery by transfection agents or CTB, we used alternative means to deliver LPS intracellularly. CL suppressed iLPS-induced LDH and IL-1β release when FuGENE (Fu), Xfect, or electroporation were used to deliver LPS in primed BMDM (Fig. EV4), confirming that CL inhibits CASP4/11 signalling regardless of the intracellular LPS delivery system. To evaluate the contribution of CASP11 in response to electroporated LPS, we primed and electroporated $Casp11^{-/-}$ BMDM in parallel. Electroporation of WT BMDM in the presence of LPS resulted in a significant increase in LDH and IL-1β release, which was reduced by CL to the level measured in control and $Casp11^{-/-}$ electroporated cells (Fig. EV4E,F). Thus, CL suppresses iLPS-induced CASP11 signalling outputs even when LPS carriers are not used, further validating CL as a bona fide CASP11 inhibitor.

To determine whether exogenously added CL relocates to the cytosol, we cultured BMDM with a modified form of CL that is covalently attached to a fluorescent probe (CL-Bodipy). We found that CL was indeed internalised by macrophages (Appendix Fig. S6). Lipid internalisation depends on cell type and liposome properties (size, charge, fluidity) and occurs via endocytosis, membrane fusion, or protein-mediated transport (Gandek et al, 2023; Nel et al, 2009; Salloum et al, 2023). LPS vesicles are taken up by macrophages via endocytosis and then escape early endosomes to interact with CASP11 (Kunsmann et al, 2015; Parker et al, 2010; Vanaja et al, 2016). To investigate whether CL vesicles follow a similar uptake route into macrophages, we used TMR-Dextran 70 kDa as a marker of endocytosis (Li et al, 2015). Co-incubation of cells with CL-bodipy plus TMR-Dextran 70 kDa for 30 min showed that CL co-localises with Dextran within macrophages (Fig. EV5). Moreover, pre-incubation of macrophages with the endocytosis inhibitors Amiloride (EIPA) or Wortmannin (Kjeken et al, 2001; Koivusalo et al, 2010) significantly reduced CL uptake by macrophages (Fig. EV5). Collectively, these data demonstrate that macrophages actively internalise extracellular CL through endocytosis.

Endocytosis is an established route for liposomal delivery to organelles (Matthaeus and Taraska, 2021; Popescu et al, 2006; Popov, 2022). A previous study showed that exogenous CL localises to mitochondria using the mitochondrial dye MitoTracker (Ikon et al, 2015), which we confirmed by showing that 18 h after exposure, CL-Bodipy co-localised with TOMM20 in macrophage mitochondria (Fig. 2C; Appendix Fig. S7). Thus, CL is taken up by cells and accumulates in the mitochondria, presumably in the mitochondrial outer membrane from which it has access to cytosolic proteins such as CASP4/11.

In the context of a bacterial infection, guanylate-binding proteins (GBP) promote LPS binding to CASP4/11 (Kirkby et al, 2023; Santos et al, 2020; Tretina et al, 2019). We sought to determine whether CL could inhibit LPS-induced noncanonical inflammasome indirectly, by targeting GBPs. Thus, we tested BMDM deficient in the chromosome 3 GBP ($Gbp^{chr3-/-}$), in which the cluster of GBPs that promote CASP11 responses to bacteria

(*Gbp1, Gbp2, Gbp3, Gbp5, Gbp7*, and *Gbp2ps*) is deleted (Enosi Tuipulotu et al, 2023; Meunier et al, 2014; Santos and Broz, 2018; Yamamoto et al, 2012). We treated Pam-primed *Gbp*chr3−/− BMDM with iLPS in the absence or presence of CL. CL suppressed iLPS-induced IL-1β, LDH, and IL-18 release in *Gbp*chr3−/− BMDM (Appendix Fig. S8). These data do not rule out the possibility that CL interacts with GBPs, but demonstrate that these GBPs are dispensable for the suppressive effect of CL on noncanonical inflammasome signalling.

Given that exogenous CL is internalised by cells and its inhibitory activities do not involve GBPs, we hypothesised that, once in the cytosol, CL may prevent CASP4/11 activation by blocking LPS interaction with CASP4/11. To test this, we incubated the cytosolic fraction of BMDM with biotinylated LPS in the absence or presence of CL (or unconjugated LPS as a control) and then pulled down biotinylated LPS using streptavidin beads. LPS pulled down CASP11 in CL-untreated cells, and this was suppressed by CL in a dose-dependent manner, similar to that of unconjugated LPS (Fig. 2D). Thus, CL prevents LPS binding to CASP11, thereby suppressing LPS-induced CASP11 activation.

This led us to hypothesise that CL may block the interaction between LPS and CASP4/11 by targeting either of these interaction partners. To test whether CL targets CASP4, we employed the DmrB dimerisation system that enables drug-inducible, LPS-independent CASP4 activation. Here, we transfected HEK293T cells with a construct encoding full-length CASP4, N-terminally fused with a V5-tagged DmrB domain (V5-DmrB-FL_CASP4). Cells were treated with the dimeriser drug AP20187 (AP) to induce DmrB-mediated dimerisation, autocleavage, and activation of CASP4. Cells were treated with AP in the presence or absence of CL, and cell lysates were assessed for CASP4 cleavage. CL inhibited AP-induced CASP4 cleavage (Fig. 2E), indicating that CL suppresses CASP4 signalling independently of LPS. We hypothesised that CL may compete with LPS for binding to CASP4/11. Given that LPS interacts with the CARD domain of CASP4/11 (J Shi et al, 2014), we reasoned that CL may target the CARD domain to block LPS interactions and resultant CASP4/11 activation. To test this, we expressed a V5-DmrB-tagged variant of CASP4 that lacks the CARD domain (V5-DmrB-ΔCARD_CASP4) in HEK293T cells and induced CASP4 activation with AP, in the absence or presence of CL. When CASP4 lacked its CARD domain, CL failed to suppress AP-induced CASP4 cleavage (Fig. 2F). For both full-length and ΔCARD-CASP4 constructs, we verified that AP-induced CASP4 cleavage represented CASP4 auto-processing activity, as this cleavage was blocked when we mutated the catalytic cysteine (C258A) (Fig. 2E,F). Together, these data suggest that CL targets the CARD of CASP4 to prevent AP-induced CASP4 dimerisation and auto-processing and thereby suppresses non-canonical inflammasome signalling.

To determine whether CL directly binds to the CASP4-CARD, we employed a non-cellular, fully recombinant system. We generated recombinant protein for the CARD domain of CASP4 tagged with the enhanced green fluorescent protein (CARD-eGFP) and incubated this with liposomes containing phosphatidylcholine (PC) or CL alone, or varying ratios of CL to PC. High-speed centrifugation then separated liposomes from unbound CARD-eGFP. CARD-eGFP levels were quantified by eGFP fluorescence in both liposomal (bound) and soluble (unbound) fractions. While liposomes containing PC alone showed no fluorescence, liposomes

containing 75 and 100% CL exhibited significant fluorescence (Fig. 2G), indicating that CL binds to the eGFP-tagged CASP4-CARD. In all, these data demonstrate that CL binds the CARD domain, and we propose this binding thereby (i) blocks LPS from binding to the CARD, and thus inhibits LPS-induced CASP4/11 activation, and (ii) sterically impedes artificial dimerisation using the AP drug - either by blocking the capacity of the AP drug to simultaneously bind to two DmrB domains, or by sterically impeding the AP/DmrB-induced dimerisation of the CARD and/ or the protease domain.

## Cardiolipin specifically inhibits CASP4/11 and not CASP1

Like CASP4/11, CASP1 also contains a CARD that is required for its activation. To test whether CL inhibits CASP1 activation, we treated primed BMDM with NLRP3/CASP1 activators (nigericin, silica) or the AIM2/CASP1 activator poly(dA:dT), and primed HMDM with nigericin, in the absence or presence of CL. CL did not affect IL-1β and LDH release induced by nigericin, silica, or poly(dA:dT) in BMDM (Fig. 3A,B), nor nigericin-induced IL-1β release in HMDM (Fig. 3C). Thus, CL specifically inhibits CASP4/11 without affecting CASP1, NLRP3, or AIM2 activity in macrophages.

Our discovery that CL is a specific CASP4/11 inhibitor offers a new tool to study CASP4/11 functions in diverse inflammatory pathways and diseases and gives proof-of-concept that targeting the CARD allows specific inhibition of CASP4/11. Such specificity may be crucial, as generic IL-1β pathway inhibition in clinical trials for several diseases increased the risk of opportunistic infections (Lopalco et al, 2016; Marshall, 2014; Salliot et al, 2009; Winthrop, 2006).

## Cardiolipin mitigates CASP11-induced pro-inflammatory IL-1β secretion in vivo

We next sought to determine whether CL blocks pathological noncanonical inflammasome responses in vivo, using a murine endotoxemia model that engages CASP11 signalling (Kayagaki et al, 2013; Napier et al, 2016). We intraperitoneally (i.p.) administered CL (25 μg/g) or HEPES (CL vehicle) immediately prior to challenge with *Pseudomonas aeruginosa* LPS (10 μg/g) or PBS (vehicle) in wild-type and *Casp11*−/− mice. TLR4- and CASP11-dependent LPS-induced inflammatory responses (IFN-γ, IL-6, TNF, and IL-1β in sera) and weight loss were measured 6 h post-injection. IFN-γ, IL-6, TNF, and IL-1β in serum and changes in weight loss were undetectable in PBS-challenged animals regardless of CL administration and were significantly increased by LPS challenge (Fig. 4). CL alone did not affect body weight, while it significantly blunted LPS-induced weight loss (Fig. 4A), supporting the previously reported safety of CL when injected i.p. in mice (Ikon et al, 2018; Ordóñez-Gutiérrez et al, 2015). LPS induced significant serum IFN-γ, IL-6, and TNF levels, which were not dependent on CASP11 and not suppressed by CL (Fig. 4B–D). These data are in line with our earlier findings that CL does not affect TLR4 activation induced by high LPS concentrations (Fig. EV3A,B and a previous study (Pizzuto et al, 2019)) and suggest that CL does not prevent TLR4-dependent priming in vivo. Further, we treated BMDM with increasing concentrations of the same LPS strain used in our in vivo model (LPS from *Pseudomonas*

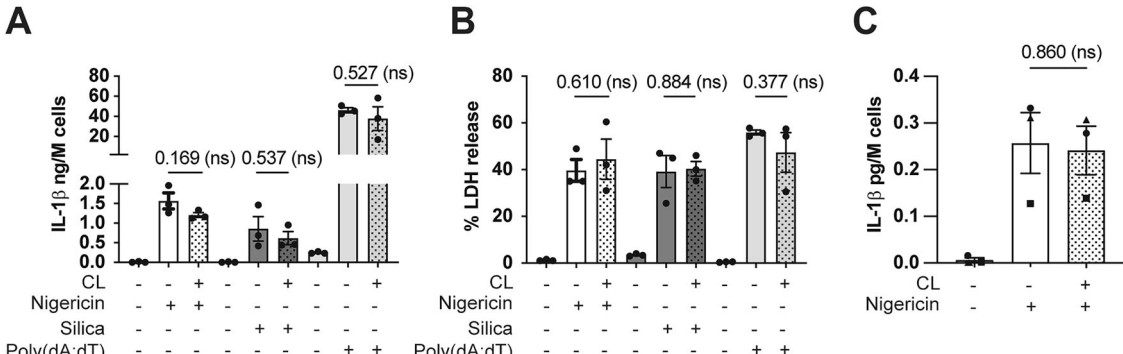

**Figure 3. Cardiolipin inhibits CASP4/11 but not CASP1 activation.**

WT BMDM (**A**, **B**) and HMDM (**C**) were primed for 4 h with 1 μg/mL Pam₃CSK₄. Cell culture medium was then replaced with 10 μM nigericin, 200 μg/mL silica, or 5 μg/ml of poly(dA:dT) in the presence of 10 μM CL (+) or HEPES vehicle (−). Cells were incubated for 1 h (nigericin), 18 h (silica), or 4 h (poly(dA:dT). Cell supernatants were analysed for cleaved IL-1β (ELISA) and LDH (cytotoxicity assay). Data information: Each symbol is the mean of technical triplicates from an independent biological replicate. Bars are the mean of three independent biological replicates (*n* = 3) ± SEM. Statistical analysis: Data were verified for normality using a Shapiro–Wilk test and analysed by unpaired *t* test. *P* values are reported above bars. Statistical significance was defined as follows: significant difference for *P* < 0.05 (*), not significant for *P* ≥ 0.05 (ns). Source data are available online for this figure.

*aeruginosa*, PA-LPS), in the presence of CL (Appendix Fig. S9). TLR4 is exquisitely sensitive to LPS; 10 ng/ml extracellular LPS induces sub-maximal signalling inhibited by 10 μM CL, while 100 ng/ml LPS induces maximal signalling and this and higher doses are not inhibited by 10 μM CL. By contrast, at 1 μg/mL, PA-LPS induced CASP11-dependent IL-1β release, which was significantly reduced by 10 μM CL (Appendix Fig. S9).

LPS induced significant levels of circulating IL-1β, which were significantly reduced in *Casp11*⁻/⁻ compared to wild-type mice, demonstrating the engagement of CASP11 noncanonical inflammasome by PA-LPS in vivo (Fig. 4E). CL significantly suppressed LPS-induced IL-1β production to levels similar to *Casp11*⁻/⁻ mice (Fig. 4E). Thus, CL suppresses CASP11 noncanonical inflammasome signalling during murine endotoxemia.

In all, our in vivo data highlight CL protective effects in suppressing endotoxemia-induced weight loss and inflammasome-driven cytokine production. Safety and efficacy make CL an attractive candidate for developing therapies to inhibit LPS/CASP4/11 signalling sequelae such as organ damage and lethality (R Chen et al, 2019; Cheng et al, 2017; Deng et al, 2018; Hagar et al, 2013; Kajiwara et al, 2014; Kayagaki et al, 2011, 2015; Tang et al, 2018; Wei et al, 2024).

## Conclusions and perspectives

In summary, this study shows that natural unsaturated CL, unlike saturated CLs and RS-LPS, is a specific inhibitor of CASP4/11 with in vivo efficacy. By identifying CL as a specific CASP4/11 inhibitor, we identify a tool reagent to study the involvement of CASP4/11 in inflammatory pathways both in vitro and in vivo. CL may also be exploited in new approaches for suppressing LPS-induced organ damage and lethality. Here, one key advantage of CL is that it does not block TLR4 or canonical inflammasome responses and is thereby unlikely to compromise these key pathways of antimicrobial defence. In a murine model of polymicrobial sepsis, mice lacking both TLR4 and CASP11 showed higher mortality than those only lacking CASP11 (Deng et al, 2018). This suggests that

preserving TLR4-dependent anti-bacterial defence while blocking CASP4/11-induced cell death and IL-1β production could offer benefit for treating sepsis. When considering CL as a therapeutic, the route of administration requires careful consideration. Intra-tracheal administration of CL following LPS exposure in a pneumonia murine model resulted in CL degradation, which produced pro-inflammatory CL metabolites and lung injury (Chakraborty et al, 2017; Ray et al, 2010). By contrast, i.p. administration of CL showed no adverse effects, by us and others (Ikon et al, 2018; Ordóñez-Gutiérrez et al, 2015), demonstrating safe in vivo application via this route. Moreover, the inability of saturated CLs and RS-LPS to dampen CASP4/11 signalling suggests that di-unsaturated chains are needed for lipids to prevent CASP4/11 activation, providing new molecular insight into the mechanism of caspase regulation by lipids that can guide the design of new inhibitors. Ultimately, the discovery that CL inhibits CASP4/11 may have important implications in future studies of noncanonical inflammasome regulation by bacteria and mitochondria. Such future studies may provide the missing links to explain why modifications of endogenous CL are associated with inflammatory disorders in diseases such as Barth Syndrome (Pizzuto and Pelegrin, 2020).

## Limitations of the study

We were unable to test the impact of CL in a TLR4-independent in vivo endotoxemia model (e.g. in *Tlr4*⁻/⁻ mice (Kayagaki et al, 2013)). Future studies should address CL actions in such a model, to definitively rule out the possibility that CL inhibits in vivo CASP11 signalling via suppressing TLR4-dependent CASP11 priming.

In addition, fluorescently labelled saturated CLs and RS-LPS are not commercially available. This limited our ability to directly compare their internalisation to that of unsaturated CLs. Because of this, we are unable to definitively rule out the unlikely possibility that these lipids were not efficiently delivered intracellularly by our transfection methods.

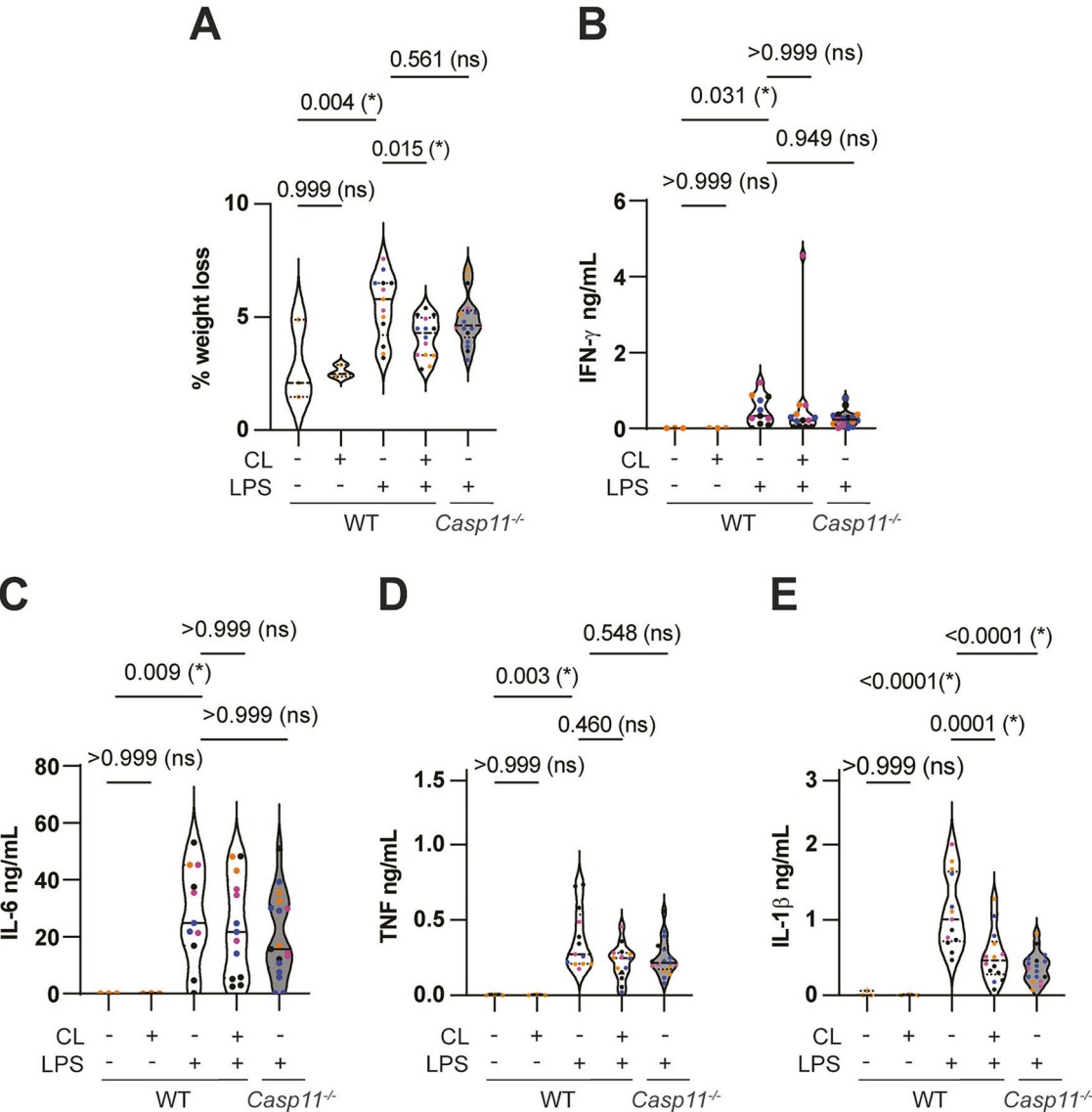

**Figure 4. Cardiolipin mitigates endotoxemia-induced systemic IL-1β in vivo.**

WT and Casp11$^{-/-}$ mice were weighed and injected intraperitoneally (i.p.) with HEPES or 25 μg/g CL. 10 min later, mice were i.p. challenged with PBS or 10 μg/g LPS. After 6 h, animal weight was recorded and blood was collected. Body weight loss was calculated as a percentage of the initial weight (**A**). IFNγ (**B**), IL-6 (**C**), TNF (**D**), and cleaved IL-1β (**E**) were quantified in sera by ELISA. Data information: Violin plot of data from four different cohorts of mice (individual mice from each cohort shown as colour-matched data points). WT PBS and WT PBS + CL $n = 3$; WT HEPES + LPS $n = 13$; WT CL + LPS $n = 13$; and Casp11$^{-/-}$ HEPES + LPS $n = 17$. Statistical analysis: Data were verified for normality using a Shapiro–Wilk test and analysed by (**A, E**) one-way ANOVA Sidak's multiple comparisons test, (**B–D**) Kruskal–Wallis with Dunn's multiple comparisons test, $P$ values are reported above bars. Statistical significance was defined as follows: significant difference for $P < 0.05$ (*), not significant for $P \geq 0.05$ (ns). Source data are available online for this figure.

# Methods

### Reagents and tools table

| Reagent/resource | Reference or source | Identifier or catalogue number |
|---|---|---|
| **Experimental models** | | |
| HEK293T | ATCC | CRL-3216 |
| Human bronchial epithelial cells HBEC-KT | ATCC | CRL-4051 |

| Reagent/resource | Reference or source | Identifier or catalogue number |
|---|---|---|
| C57BL/6J mice WT | The Jackson Laboratory | RRID:IMSR_JAX:000664 |
| C57BL/6J mice Nlrp3$^{-/-}$, | The Jackson Laboratory And Tschopp Lab (University of Lausanne, Switzerland) | RRID:IMSR_JAX:021302 And (Martinon et al, 2006) |

| Reagent/resource | Reference or source | Identifier or catalogue number |
|---|---|---|
| C57BL/6J mice Casp11⁻/⁻ | The Jackson Laboratory | RRID:IMSR_JAX:024698 |
| C57BL/6J mice Tlr4⁻/⁻ | Prof Matthew Sweet lab (Institute for Molecular Bioscience) | Curson et al, 2023; Hoshino et al, 1999 |
| C57BL/6ncrlanu mice and Gbp^chr3⁻/⁻ | Prof Yamamoto (Osaka University, Japan) | Yamamoto et al, 2012 |
| Human blood buffy coats | Australian Red Cross Blood Service | N/A |
| **Recombinant DNA** | | |
| pEF6 expression vector | Thermo Fisher Scientific | K961020 |
| **Antibodies** | | |
| V5 antibody | | |
| Anti-murine Caspase-1 (p20) mab (Casper-1), raised in mouse | Adipogen Life Sciences | AG-20B-0042-C100 |
| Anti-human GSDMD antibody, raised in rabbit | Causabio | CSB-PA009956GA01HU |
| Anti-Caspase-11 clone17d, raised in rat | Novus or Merck | NOVNB12010454 or C1354 |
| Anti-Caspase-11, raised in rabbit | Abcam | Ab180673 |
| Anti-Caspase-4, raised in mouse | Santa Cruz or Proteintech | 56056 or 67398-1-Ig |
| Anti-V5-tag Sv5-PK1 | Abcam | Ab27671 |
| Anti-murine/human IL-1beta, raised in goat | Rnd System | AF-401-NA |
| Anti-Mouse IgG HRP-conjugated, raised in horse | Cell Signalling | 7076S |
| Anti-Rabbit IgG HRP-conjugated, raised in goat | Cell Signalling | 7074S |
| Anti-Goat IgG HRP-conjugated, raised in monkey | Abacus | JI705035147 |
| Anti-Rat IgG HRP-conjugated, raised in goat | Cell Signalling | 7077S |
| Anti-Tubulin Rhodamine-conjugated hFab | Biorad | 12004165 |
| Anti-Rabbit IgG-Star Bright B700 raised in goat | BioRad | 12004161 |
| Anti-TOMM20, raised in rabbit | Abcam | Ab186735 |
| Anti-rabbit Alexa Fluor-594 | Molecular Probes | A21442 |
| **Oligonucleotides and other sequence-based reagents** | | |
| **Chemicals, enzymes, and other reagents** | | |
| Unsaturated cardiolipin 18:2 | Avanti Polar Lipid | 840012P |
| Unsaturated cardiolipin 18:2 | Merck | C0563 |
| Saturated cardiolipin 16:0 | Avanti Polar Lipid | 710333P |
| Saturated cardiolipin 18:0 | Avanti Polar Lipid | 710334P |

| Reagent/resource | Reference or source | Identifier or catalogue number |
|---|---|---|
| Topfluor® cardiolipin | Avanti Polar Lipid | 810286 |
| HEPES | Gibco | 15630080 |
| Chloroform (chcl3) | Merck | 650498 |
| Poly(da:dt) | Invivogen | Tlrl-patn-1 |
| Ultrapure LPS from Escherichia coli O111:B4 | Invivogen | Tlrl-3pelps |
| Ultrapure LPS from Escherichia coli O111:B4 | Enzo Life Science | ALX-581-014-L002 |
| Ultrapure LPS from Escherichia coli K12 | Invivogen | Tlrl-peklps |
| Ultrapure LPS from Salmonella enterica serovar Minnesota mutant R595 | Invivogen | Tlrl-smlps |
| Ultrapure LPS from Rhodobacter spheroides | Invivogen | Tlrl-prslps |
| LPS from Pseudomonas aeruginosa | Merck | L8643 |
| Fugene®HD | Promega | E2311 |
| Lipofectamine™ LTX Reagent with PLUS™ Reagent (LTX) | Invitrogen | A12621 |
| Xfect | Clontech Laboratories | 631318 |
| Cholera toxin B (CTB) | Merck | C9903 |
| Opti-modified Eagle's medium (MEM)™ reduced-serum medium (optimem) | Invitrogen | 31985-070 |
| OptiMEM™ | Gibco/Thermo Fisher Scientific | 51985091 |
| Dulbecco's modified Eagle's medium (DMEM) F12 | Biowest | 11320 |
| IKratinocyte medium | Gibco | 17005042 |
| RPMI 1640 | Sigma-Aldrich/ Merck or Gibco | R8758 Or 11875093 |
| Foetal bovine serum premium | Biowest | A5256701 |
| Penicillin–streptomycin | Gibco/Life Technologies | 15140122 |
| Penicillin–streptomycin | Lonza | 17-603 DE17-603 |
| L-glutamine | Lonza | BEBP17-605E |
| Glutamax | Gibco/Life Technologies | 35050061 |
| Recombinant human macrophage colony-stimulating factor CSF-1 (endotoxin-free) | Expressed and purified by the University of Queensland Protein Expression Facility | N/A |
| Dulbecco's phosphate-buffered saline (PBS) | Thermo Fisher Scientific | J67670.K2 |
| Pam3CSK4 | Invivogen | Tlrl-pms |
| All Blue Protein Ladder | Biorad | 1610393 |
| Protease inhibitor Cocktail | Sigma-Aldrich Roche | 11836170001 |

| Reagent/resource | Reference or source | Identifier or catalogue number |
|---|---|---|
| Benzonase | Sigma-Aldrich | E1014 |
| Dithiothreitol (DTT) | Thermo Fisher Scientific | 10708984001 |
| Nupage™ LDS Sample Buffer | Thermo Fisher Scientific | NP0008 |
| 20% SDS Solution | Biorad | 1610418 |
| VX765 | Invivogen | Inh-vx765i-1 |
| MCC950 | Medchemexpress | HY-12815 |
| Sodium Chloride nacl | Sigma-Aldrich | S3014 |
| Triton X-100 | Sigma-Aldrich | 648463 |
| Tris-hcl | Thermo Fisher Scientific | 15568025 |
| Supersignal™ West Femto Maximum Sensitivity Substrate | Thermo Fisher Scientific | 34094 |
| Histopaque-1077 | Sigma-Aldrich/ Merck | 10771 |
| Sepmate™ PBMC isolation tubes | STEMCELL™ | 85450 |
| Lipofectamine 2000 | Thermo Fisher | 11668027 |
| Dimeriser drug AP20187 B/B Homodimerizer | Medchemexpress | HY-13992 |
| Complete keratinocyte medium | Gibco | 17005042 |
| Streptavidin magnetic beads | Promega | Z5481 |
| Ficoll-Paque Plus | Merck—GE Healthcare | GE17-1440-02 |
| Iscove's Modified Dulbecco's Medium (IMDM) | Gibco | 12440053 |
| Clearcoli BL21(DE3) | Lubio Science-Astral Scientific | 60810-1 |
| Isopropyl-b-D-galactopyranoside (IPTG) | Bioline BIO-37036 | BIO-37036 |
| 1 M Tris-HCl | Thermo Fisher Scientific | 15568025 |
| 5 M NaCl | Thermo Fisher Scientific | AM9759 |
| Imidazole | Merck | I202 |
| B-mercaptoethanol | Thermo Fisher Scientific | 21985023 |
| Tween-20 | Sigma-Aldrich | P1379-1L |
| Ni Sepharose® 6Fast Flow | Cytivage Healthcare - Bio-Strategy | 17-5318-02 |
| Glycerol | Chem-Supply | GA010-500ml |
| 10 M NaOH | Sigma-Aldrich | 72068-100 ML |
| Dithiothreitol | Sigma-Aldrich | D9779-10G |
| L-alpha-phosphatidylcholine (PC) | Avanti Polar Lipid | 840051 |
| Pierce™ 16% Formaldehyde (PFA) | Pierce/Thermo Scientific™ | 28906 |
| Phalloidin-ifluor 594 | Abcam | A176757 |

| Reagent/resource | Reference or source | Identifier or catalogue number |
|---|---|---|
| Alexa Fluor 647 Phalloidin | Thermo Fisher Scientific | A22287 |
| DAPI | Sigma-Aldrich | D9542 |
| Prolong Gold Antifade Reagent | Thermo Fisher Scientific | P36934 |
| TMR-Dextran 70 kDa | Thermo Fisher Scientific | D1819 |
| Amiloride | Sigma-Aldrich | A3085 |
| Wortmannin | Selleck Chem | S2758 |
| Mycoprobe Mycoplasma Detection Kit | R&D Systems | CUL001B |
| Murine TNF duoset ELISA Kit | R&D Systems | DY410 |
| Murine IL-1β duoset ELISA Kit | R&D Systems | DY401 |
| Murine IL-6 duoset ELISA kit | R&D Systems | DY406 |
| Murine IFNγ duoset ELISA kit | R&D Systems | DY485 |
| Murine TNF ELISA™ Kit | Invitrogen | Cat #88-7324-88 |
| Murine IL-1β ELISA™ Kit | Invitrogen | 88-7013A-88 |
| Human IL-1β ELISA™ Kit | Invitrogen | KHC0011 |
| Murine IL-18 ELISA™ Kit | Invitrogen | BMS618-3TEN |
| Human IL-18 duoset ELISA™ Kit | R&D Systems | DY318 |
| Human TNF duoset ELISA™ Kit | R&D Systems | DY210 |
| Human IL-1β duoset ELISA™ Kit | R&D Systems | DY201 |
| Cytotoxicity Detection Kit (LDH) | Merck | 11644793001 |
| Cytox96 non-radioactive cytotoxicity assay | Promega | G1780 |
| RNA purification rneasy kit | Qiagen | 74104 |
| Reverse transcription iscripttm cdna Synthesis kits | Biorad | 1708890 |
| Magnetic-assisted CD14+ cell sorting | Miltenyi Biotec | 130-097-052 |
| **Software** | | |
| GraphPad Prism 10 | GraphPad Software, Inc. | |
| Image Lab Version 6.1 | Biorad | |
| ImageJ Version 2.1.0 | NIH, open source image processing | |
| Zeiss Zen 2012 Black software | Zeiss | |
| **Other** | | |
| Ultrasonic bath XUBA1 | Grant | 144628 |
| Synergy HT Microplate Readers | Biotek | B-SHT |
| Tecan Microplate Reader | Tecan | 30190085 |
| Nanodrop 2000 | Thermo Fisher Scientific | ND-2000 |
| Chemidoc MP Imaging System | Biorad | 12003154 |
| Trans-Blot Turbo Transfer System | Biorad | 17001915 |

| Reagent/resource | Reference or source | Identifier or catalogue number |
|---|---|---|
| Mini-PROTEAN® TGX™ Precast Gels 4–20% | Biorad | 4561093 |
| Trans-Blot Turbo RTA Midi 0.2 µm Nitrocellulose Transfer Kit | Biorad | 1704271 |
| Dynamag-2 magnetic rack | Invitrogen | 12321D |
| Microplate reader Cytation5 | Biotek | |
| Zeiss Axiovert 200 Inverted Microscope Stand with LSM880 Confocal Scanner and Fast Airyscan Detector | Zeiss | |

## Methods and protocols

### Cardiolipin liposome preparation

The cardiolipin used in this work is di-unsaturated cardiolipin (18:2) extracted from bovine heart, purchased from Avanti Polar Lipid (840012P) or Merck (C0563), saturated 16:0 CL (710333P) and 18:0 CL (710334P), and 18:2 CL-Bodipy purchased from Avanti Polar Lipid (TopFluor® Cardiolipin 810286). CL was dissolved in $CHCl_3$ (Merck, 650498) at 1 mg/mL, and lipid films were prepared by solvent evaporation under a filtered nitrogen stream before being dried overnight and stored at $-20\,°C$. Before each experiment, liposomes were freshly formed by resuspending lipid films with filtered 10 mM HEPES (Gibco, 15630080), heating for 20 min at $70\,°C$, and then sonicating for 5 min at 44 kHz in an ultrasonicator bath (Grant XUBA1).

### Intracellular delivery of LPS or poly(dA:dT)

Poly(dA:dT) (InvivoGen, tlrl-patn-1) or ultrapure LPS from *Escherichia coli* 0111:B4 (InvivoGen, tlrl-3pelps or ALX-581-014-L002, Enzo Life Science), *Escherichia coli* K12 (InvivoGen, tlrl-peklps), *Salmonella enterica serovar minnesota mutant* R595 (InvivoGen, tlrl-smlps), *Pseudomonas aeruginosa* (Merck L8643), or *Rhodobacter spheroides* (InvivoGen, tlrl-prslps) were first mixed with FuGENE®HD (Promega, E2311), Lipofectamine™ LTX Reagent with PLUS™ Reagent (LTX) (Invitrogen, A12621), Xfect (Clontech Laboratories, 631318), or cholera toxin B (CTB) (Merck, C9903) in a small volume (1/10$^{th}$ final volume) of Opti-modified Eagle's medium (MEM)™ reduced-serum medium (OptiMEM) (Invitrogen, 31985-070,) that was previously heated to $37\,°C$. The mix was vortexed, and then incubated for 15 min at room temperature to allow complexes to form, and then added to cells in OptiMEM™. Final concentrations are indicated in the figure legends.

### HEK293T cell culture, transfection and treatments

All cells were cultured in humidified incubators at $37\,°C$ and with 5% $CO_2$. HEK293T (ATCC CRL-3216) in Dulbecco's modified Eagle's medium (DMEM; Gibco) supplemented with 10% heat-inactivated foetal bovine serum (FBS) and 1% penicillin–streptomycin (Pen-Strep).

The full-length coding sequence of human *CASP4* (residues 1–377) and delta-CARD *CASP4* (residues 81–377) were cloned as N-terminal DmrB fusions into the mammalian pEF6 expression vector (Invitrogen), as either the wild-type sequence or with an inactivating mutation in the catalytic cysteine (C258A). Sequences were cloned in-frame with an N-terminal V5 tag and a C-terminal HA tag.

The DmrB constructs were transfected into HEK293T cells seeded in 10-cm cell culture dishes using Lipofectamine 2000 (Thermo Fisher). After overnight expression, the cells were harvested and re-seeded at $0.3 \times 10^6$ cells per well in 24-well plates in OptiMEM. After 3 h, 30 µM CL was added (or HEPES vehicle), and 1 h later OptiMEM or the dimeriser drug AP20187 (AP, 1 µM) was added. 30 min later, cell lysates were collected, and CASP4 expression and cleavage were assessed by western blot using the V5 antibody.

### HBEC cell line culture and treatments

Immortalised human bronchial epithelial cells HBEC-KT (ATCC CRL-4051) were maintained in complete keratinocyte medium (Gibco 17005042, supplemented with the provided bovine pituitary enzyme and epidermal growth factor) and passaged at 70–80% confluency. Cells were seeded at $0.1 \times 10^6$ cells in 100 µL per well in 96-well plates in complete keratinocyte medium supplemented with 1 µg/mL $Pam_3CSK_4$. After 18 h, the cell culture medium was replaced with OptiMEM alone or containing 10 µM of the NLRP3 inhibitor MCC950 or the CASP1/4 inhibitor VX765 (MedChem Express). One hour later, OptiMEM or 2 µg/mL of LPS complexed with 0.25% LTX were added in the absence or presence of 10 µM CL. Four hours later, supernatants were collected, centrifuged at $600\times g$, and assayed for cytokine and LDH release.

### Differentiation of bone marrow-derived macrophages (BMDM)

Experiments conducted at the Biomedical Research Institute of Murcia used wild-type or $Nlrp3^{-/-}$ C57BL/6J male and female mice between 8 and 13 weeks of age and bred under specific pathogen-free (SPF) conditions, in accordance with the Hospital Clínico Universitario Virgen Arrixaca animal experimentation guidelines, and the Spanish national (RD 1201/2005 and Law 32/2007) and EU (86/609/EEC and 2010/63/EU) legislation. Accordingly, no specific procedure approval is needed when animals are sacrificed to obtain biological material. Mice were euthanised by $CO_2$ inhalation and the bone marrow was flushed from the leg bone cavity and resuspended in differentiation medium (DMEM medium, with L-glutamine, without sodium pyruvate (Biowest, 11320) supplemented with 10% heat-inactivated foetal bovine serum premium (Biowest, A5256701), 2 mM glutamine (Lonza, BEBP17-605E), 50 U/mL penicillin, 50 µg/mL streptomycin (PEN-STREP, Lonza, 17-603 DE17-603), and 20% of supernatant from L929 cultures. The bone marrow cell suspension was maintained in Petri dishes in a $37\,°C/5\%\ CO_2$ atmosphere. After 2 days, the differentiation medium was supplemented, and cells were maintained for 4 extra days, before replating in DMEM medium, with L-glutamine, without sodium pyruvate, supplemented with 10% heat-inactivated foetal bovine serum premium (Biowest, A5256701).

Alternatively, experiments at the Institute for Molecular Bioscience (University of Queensland) used wild-type, $Tlr4^{-/-}$, $Casp11^{-/-}$, and $Nlrp3^{-/-}$ C57BL/6J male and female mice between 6 and 14 weeks of age and bred under specific pathogen-free (SPF) facilities at the University of Queensland. All protocols involving mice were approved by the University of Queensland Animal Ethics Committee, and compliance with relevant ethical regulations was ensured (2023/AE000019, 2023/AE000020, 2020/AE000419). Mice

were euthanised by $CO_2$ inhalation, the bone marrow was flushed from the bone cavity, filtered, centrifuged at $400 \times g$ for 5 min and resuspended in differentiation medium consisting of RPMI 1640 medium (Life Technologies, 11875093) supplemented with 10% heat-inactivated and endotoxin-free foetal calf serum (FCS) (Gibco), 2 mM GlutaMAX (Life Technologies, 35050061), 50 U per ml penicillin–streptomycin (Life Technologies, 15140122) and 150 ng/ml recombinant human macrophage colony-stimulating factor (CSF-1; endotoxin-free, expressed and purified by the University of Queensland Protein Expression Facility). After 5 days, the differentiation medium was supplemented, and cells were maintained for a further day before replating for experiments in the cell differentiation medium.

Experiments conducted at the Australian National University (Xfect/LPS/polydAdT transfection) used C57BL/6NCrlAnu mice and *Gbp*$^{chr3-/-}$ mice (Yamamoto et al, 2012) sourced from Osaka University. Primary BMDM were differentiated and cultured in Dulbecco's Modified Eagle Medium (DMEM) (11995073, Gibco Thermo Fisher Scientific) with 30% L929-conditioned medium 1% penicillin and streptomycin (10378016, Gibco Thermo Fisher Scientific) and 10% foetal bovine serum (FBS; F8192, Sigma).

### BMDM stimulation

BMDM differentiated for 6 days were washed and harvested using Dulbecco's phosphate-buffered saline (PBS) (Thermo Fisher Scientific, J67670.K2). The cells were then counted, centrifuged for 5 min at $500 \times g$ and resuspended in full medium to a concentration of $1 \times 10^6$ cells/mL and distributed in 96-well plates (100 μL/well), 24-well plates (500 μL/well), or six-well plates (2 mL/well) and cultured overnight. Medium was added, either alone or supplemented with 1 μg/mL $Pam_3CSK_4$ (InvivoGen, tlrl-pms). After 4 h, the cell culture medium was replaced with OptiMEM alone or containing 10 μM of the NLRP3 inhibitor MCC950, or the CASP1/4 inhibitor VX765 (MedChem Express) 1 h later, OptiMEM or LPS complexed with CTB were added in the absence or presence of 10 μM CL. Eighteen hours later, lysates and supernatants were collected, centrifuged at $600 \times g$, and assayed for cytokine and LDH release.

### Macrophage CASP11-LPS-binding assay

BMDM were primed for 4 h with 1 μg/mL of $Pam_3CSK_4$, then lysed in 100 μL lysis buffer (50 mM HEPES pH 7.4, 150 mM NaCl, 2 mM EGTA, 10% glycerol, and 1% Triton X-100, plus protease inhibitor 100 μL/mL) per million cells. Lysates were spun at 13,000 g for 10 min, and the pellet containing debris and non-lysed cells was discarded. The lysate was aliquoted (250 μL per tube), and reagents added to the same final volume: CL (0, 100, or 200 μg), or LPS (0 or 100 μg). Tubes were incubated at room temperature for 2 h on a Benchmark rotating wheel, and then biotinylated LPS (0 or 1 μg/mL) was added. One hour later, 6% vol/vol of pre-washed and blocked Promega Z5481 streptavidin magnetic beads were added. Beads were pre-washed with BSA-T buffer (0.15% Tween, 5% bovine serum albumin in PBS) and blocked by incubation for 1 h in BSA-T buffer on a rotating wheel. Samples were incubated with beads overnight at 4 °C. Then, supernatants (unbound fractions) were separated from beads. Beads were recovered and washed four times for 5 min in BSA-T buffer at room temperature on the rotating wheel. Then beads were recovered, and bound samples were eluted from beads by adding 40 μL of lysis buffer diluted 3:4 in

NuPAGE 4× (final NuPAGE concentration 1×) and boiling at 100 °C for 5 min. Beads were recovered from preservative solution, buffer or samples by gentle magnetic separation using a DynaMag-2 magnetic rack (Invitrogen). Unbound fractions were diluted 3:4 in NuPAGE 4× (final NuPAGE concentration 1×) and boiled at 100 °C for 5 min. The amount of CASP11 in LPS-bound and unbound fractions was assessed by western blot.

### Differentiation of human macrophages from human monocytes (HMDM)

Studies using primary human cells were approved by the UQ Human Research Ethics Committee (HE000413). The Australian Red Cross Blood Service provided buffy coats from anonymous, informed and consenting adults for this research study. Human monocytes were isolated from screened buffy coats by density centrifugation with Ficoll-Paque Plus (GE Healthcare) followed by Miltenyi Biotec magnetic-assisted CD14+ cell sorting, according to manufacturer protocols. Monocytes were differentiated to macrophages by 6 days of culture at 37 °C with 5% $CO_2$ in Iscove's Modified Dulbecco's Medium (IMDM; Gibco) medium supplemented with 10% endotoxin-free heat-inactivated foetal bovine serum (Gibco), 1% penicillin–streptomycin, 1× GlutaMAX, and recombinant human CSF-1 (150 ng/mL; endotoxin-free, produced in insect cells by the UQ Protein Expression Facility).

### HMDM stimulations

HMDM differentiated for 6 days were washed and detached from their dishes using Dulbecco's phosphate-buffered saline (PBS) (Thermo Fisher Scientific, J67670.K2). The cells were then counted, centrifuged for 5 min at $500 \times g$, and resuspended in full medium to a concentration of $0.5 \times 10^6$ cells/mL and distributed in 96-well plates (100 μL/well) or 12-well plates (1 mL/well) and cultured overnight. Medium was replaced with fresh medium, either alone or supplemented with 1 μg/mL $Pam_3CSK_4$ (InvivoGen, tlrl-pms). After 4 h, the cell culture medium was replaced with OptiMEM alone or containing 10 μM of the NLRP3 inhibitor MCC950 or the CASP1/4 inhibitor VX765 (MedChem Express). One hour later, OptiMEM or LPS complexed with LTX were added in the absence or presence of 10 μM CL. Four hours later, lysates and supernatants were collected, centrifuged at $600 \times g$, and assayed for cytokine and LDH release.

### Cytokine assays

Murine IL-18 was quantified with the ELISA™ Kit (Thermo Fisher Scientific, BMS618-3TEN). Murine TNF and IL-1β were quantified in cell supernatants using ELISA (DuoSet R&D Systems: DY401, D410 or Invitrogen: 88-7324-88, 88-7013A-88). Human TNF, IL-1β, and IL-18 were quantified in cell supernatants using the DuoSet ELISA Kit from R&D Systems (DY210, DY201 and DY318).

Absorbance was read with a Tecan Microplate Reader or a BioTek Synergy HT Microplate Reader. For in vitro experiments, cytokine amounts were reported as ng or pg per million cells to standardise the difference in cell amount/volume of media ratio between the different plate layouts.

### LDH assay

LDH activity was quantified in cell supernatants using the Cytotoxicity Detection Kit (Merck), following the manufacturer's instructions. Absorbances at 492 and 620 nm were measured with a

BioTek Synergy HT Microplate Reader every minute for 20 min, with the slopes of increase in absorbance calculated with respect to time, and background values subtracted from the value of each supernatant reported as percentages of the sum of the value measured in the supernatant and the lysate of the untreated condition (total LDH). Untreated cell lysates were also obtained to estimate total cellular LDH, and were prepared as follows: cells were lysed with 2% Triton lysis buffer comprising 150 mM NaCl (Sigma-Aldrich), 2% Triton X-100 (Sigma-Aldrich), and 50 mM Tris-HCl pH8 (Thermo Fisher Scientific) supplemented with 100 µL/mL of protease inhibitor (Sigma-Aldrich). Cells were harvested by scraping in cold lysis buffer on ice. Lysates were then incubated for 30 min on ice with a vortex every 10 min, before centrifugation for 10 min at $13,000 \times g$ in a microcentrifuge (1–14 K, Sigma) to remove cell debris.

Alternatively, LDH activity was quantified in all cell supernatants using the Cytox96 non-radioactive cytotoxicity assay (Promega). Absorbances at 492 and 620 nm were measured with a Tecan Microplate Reader and reported as percentages of the value measured in the supernatant of cells treated with 0.1% Triton for 10 min (total LDH).

### Western blotting

Cells were lysed in complete lysis buffer (66 mM Tris-Cl pH 7.4, 2% SDS, 100 mM DTT, Benzonase 0.01% vol/vol, NuPAGE 1X). Cell supernatants were concentrated following CHCl$_3$/MeOH precipitation as described earlier (Groß, 2011) and resuspended in complete lysis buffer. Samples were incubated for 5 min at 100 °C, then resolved by SDS–PAGE using 4–20% Mini-PROTEAN TGX stain-free gels (BioRad) and transferred onto nitrocellulose membrane using the Trans-Blot Turbo transfer system (BioRad). Membranes were blocked in 5% (wt/vol) dried milk in TBS-T (10 mM Tris/HCl, pH 8, 150 mM NaCl, 0.05% vol/vol Tween-20) for 1 h at room temperature. Membranes were incubated for 18 h at 4 °C with primary antibody diluted in 5% (wt/vol) dried milk in TBS-T and then 1 h at room temperature with the appropriate secondary antibody diluted in 5% (wt/vol) dried milk in TBS-T for 1 h. Membranes were developed using SuperSignal West Femto Maximum Sensitivity Substrate, ultra-sensitive enhanced chemiluminescent (ECL) (Thermo Scientific). Membranes were then visualised using a ChemiDoc MP Imaging System with Image Lab 6.1 (BioRad). Horseradish peroxidase (HRP)-conjugated secondary antibodies on membranes were inactivated by incubation with 30% hydrogen peroxide for 20 min before re-probing. The following primary antibodies were used at 1:1000: anti-human/murine IL-1β (RnD AF-401-NA), anti-human CASP1 (AG-20B-0042, Adipogen), anti-murine CASP1, anti-murine GSDMD (Ab209845, Abcam), anti-human GSDMD (Causabio, CSB-PA009956GA01HU), anti-V5-tag Sv5-PK1 (ab27671, Abcam), anti-CASP11 (Abcam ab180673, NOVUS NOVNB12010454 or Merck C1354), and anti-CASP4 (56056, Santa Cruz or Proteintech 67398-1-Ig). Secondary antibodies used were anti-rabbit IgG or anti-mouse IgG HRP-conjugated (7074S, 7076S; both Cell Signalling Technology) diluted 1:5000, anti-goat IgG HRP-conjugated (JI705035147 Abacus), or anti-Rabbit IgG-Star Bright B700 (BioRad 12004161) diluted 1:10,000. Tubulin was blotted using Rhodamine-conjugated anti-tubulin (BioRad) diluted 1:10,000 protected from light.

### Expression and isolation of recombinant CASP4-CARD

The CASP4-CARD domain (1–80 aa), codon-optimised for *E. coli*, was expressed with C-terminal eGFP-6xHis tag from pET28b vector in ClearColi BL21(DE3) (Lubio Science), grown in Luria-Bertani medium. Expression was induced at an OD$_{600}$ of 0.5 with 0.2 mM isopropyl-β-D-galactopyranoside (IPTG) at 18 °C, overnight. For purification, all glassware was first rinsed with 1 M NaOH to avoid contamination by lipopolysaccharide. Harvested bacteria were resuspended in IMAC-A buffer (20 mM Tris-HCl pH 7.9, 300 mM NaCl, 20 mM imidazole, 5 mM β-mercaptoethanol, 1% Tween-20) and lysed by sonication. The CASP4-CARD-GFP-His was purified by IMAC affinity chromatography on Ni Sepharose® 6Fast Flow (Cytiva), eluted using 300 mM imidazole in IMAC-B buffer (20 mM Tris-HCl pH 7.9, 300 mM NaCl, 300 mM imidazole). Finally, monomeric CASP4-CARD-eGFP-His was further purified to homogeneity by size-exclusion chromatography, using Superdex-75 10/300 equilibrated in SEC buffer (20 mM HEPES-NaOH pH 7.5, 300 mM NaCl, 10% glycerol, 2 mM dithiothreitol) and frozen in liquid nitrogen.

### Liposome co-sedimentation assay for CASP4-CARD binding assay

To analyse the binding of CASP4-CARD to CL, 1 µM recombinant CARD in binding buffer (20 mM HEPES-KOH pH 7.5, 150 mM KCl) was mixed with liposomes containing 1.67 mM total lipid, with increasing proportions of CL (0 to 100% mol/mol) and decreasing proportions of PC (L-alpha-phosphatidylcholine, egg, chicken, Avanti Polar Lipids), and incubated at 37 °C for 1 h with constant shaking. Subsequently, mixtures were centrifuged at $20,000 \times g$ at 4 °C for 30 min. Liposome-containing pellets were resuspended in binding buffer, using a volume equal to the supernatant volume. The GFP fluorescence (ex: 485 nm/em: 528 nm) was measured, using the microplate reader Cytation5 (Biotek).

### Immunofluorescence microscopy

BMDM or HMDM were plated on 1' glass coverslips at $1 \times 10^5$ cells per well in a 24-well plate, primed with Pam$_3$CSK$_4$ for 4 h and incubated with CL liposomes containing 1% (w/w) CL-Bodipy (TopFluor® Cardiolipin Avanti Polar Lipid 810286), in the absence or presence of 100 µg/mL 70 kDa TMR-dextran (Thermo Fisher Scientific #D1819) for 30 min or 18 h. To assess the effect of endocytosis inhibitors, BMDM were incubated with Amiloride (Sigma-Aldrich #A3085) or Wortmannin (Selleck Chem #S2758) 1 h before adding CL-Bodipy. After treatment, BMDM were washed four times and fixed in 4% PFA (Pierce) for 15 min at 37 °C. Cells were permeabilised in 0.1% saponin (Sigma-Aldrich) and non-specific binding was blocked using 0.5% BSA (Sigma-Aldrich) for 30 min, followed by incubation with primary anti-TOMM20 (1:400, Abcam ab186735) and secondary anti-rabbit Alexa Fluor-594 (A21442, Molecular Probes) or the actin probe phalloidin-594 (6.6 µM, Invitrogen), together with DAPI (0.1 µg/ml, Sigma-Aldrich) for 1 h. Coverslips were mounted with ProLong Gold Antifade Reagent (Invitrogen). Images were acquired on a Zeiss Axiovert 200 Inverted Microscope Stand with LSM880 Confocal Scanner running Zeiss Zen 2012 Black software. The microscope was equipped with 405, argon ion, and 561 lasers. A Plan Apochromat 60× (NA 1.4) oil immersion objective was used, and the Fast Airyscan Detector was employed. Images were processed in Fiji (NIH).

### Endotoxemia in vivo model

Female wild-type and *Casp11*$^{-/-}$ C57BL/6J mice between 8 and 10 weeks of age were weighed and injected intraperitoneally with 10 mM HEPES or 25 µg/g CL liposomes diluted in 10 mM HEPES. Ten minutes later, mice were injected with PBS or 10 µg/g of LPS from *Pseudomonas aeruginosa* (Sigma-Aldrich, L8643) diluted in PBS. 6 h later, mice were weighed and humanely euthanised using $CO_2$, and blood was collected by cardiac puncture. Blood was left to coagulate for 3 h at RT, and serum was collected by centrifugation at $700 \times g$ for 10 min. Serum was assayed for circulating cytokines by IFNγ, IL-6, TNF, and IL-1β ELISA.

### Quantification and statistical analysis

Statistical details of experiments, including the statistical tests used and the exact value of *n*, can be found in the figures and figure legends. For in vitro experiments, n represents the number of independent experiments (biological replicates). For in vivo experiments, *n* represents the number of mice per phenotype. In graphs, each symbol represents the mean value of technical triplicates from an independent experiment, while each bar represents the mean value across independent biological replicates, with the error bars representing the standard error of the mean (SEM) of three or more independent experiments, as indicated.

Shapiro–Wilk tests were performed to assess whether data were normally distributed. Normally distributed data were analysed using the following parametric tests: one-sample *t* test, unpaired *t* test, one-way analysis of variance (ANOVA). Data that were not normally distributed were analysed using non-parametric tests: Mann–Whitney, Kruskal–Wallis and mixed-effect analysis. Analyses are indicated in the figure legends.

*P* values are reported in the graphs and considered significant if $P < 0.05$. Nonlinear regression analysis was carried out using GraphPad Prism 10 software (GraphPad Software, Inc.). Prism 10 was also used to generate graphs, calculate SEM, and perform statistical analysis.

No blinding procedures were implemented in this study.

### Declaration of generative AI and AI-assisted technologies in the writing process

During the preparation of this work, the authors used Grammarly and the language model provided by OpenAI for assistance in proofreading, grammar checking, and identifying synonyms. After using this tool/service, the authors reviewed and edited the content and take full responsibility for the content of the publication.

## Data availability

This study includes no data deposited in external repositories.

The source data of this paper are collected in the following database record: biostudies:S-SCDT-10_1038-S44318-025-00507-z.

## Peer review information

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

## Acknowledgements

This work was supported by the Spanish Ministry of Science and Innovation (Grant PID2023-147531OB-I00, CNS2022-135105, MCIN/AEI/10.13039/501100011033 and PID2020-116709RB-I00 to PP and *Juan de la Cierva-Formación* postdoctoral fellowship FJC2018-036217-I to MP), the *Fundación Séneca* (grants 21897/PI/22, 20859/PI/18, 21081/PDC/19 and 0003/COVI/20 to PP), the Instituto Salud Carlos III (Grant AC22/00009 to PP), the European Research Council (grants ERC-2013-CoG 614578 and ERC-2019-PoC 899636 to PP), the Australian Research Council (Discovery Project DP190102285 to KS), the National Health and Medical Research Council of Australia (Fellowship 2009075 to KS; Synergy Grant 2009677 to KS), by the Barth Syndrome Foundation (BTHS Idea Grant to MP and KS), and by the *Fond National de la Recherche Scientifique* (postdoctoral fellowship CR 32774874 to MP), The John Curtin School of Medical Research PhD Scholarship (to MK), a CSL Centenary Fellowship (to SMM). Swiss National Science Foundation Postdoc Mobility Fellowship (P2BEP3) and Novartis Foundation for Biomedical Research Fellowship (21C133) (to SSB). The authors gratefully acknowledge Dr. James Curson and Prof Matthew Sweet (IMB, Brisbane, Australia) for providing BMDM from *Tlr4*$^{-/-}$ mice, Tyron Esposito and Dr. Xiaohui Wang (IMB, Brisbane, Australia) for help with HMDM processing, and Dr. Madhavi Maddugoda (IMB, Brisbane, Australia) for editing this manuscript. We thank Maria del Carmen Baños and Ana Isabel Gomez (IMIB, Murcia, Spain) and Chinh Ngo (ANU, Canberra, Australia) for technical assistance with cell culture and plasmids.

## Author contributions

**Malvina Pizzuto**: Conceptualisation; Formal analysis; Supervision; Funding acquisition; Validation; Investigation; Visualisation; Methodology; Writing—original draft; Project administration; Writing—review and editing. **Mercedes Monteleone**: Formal analysis; Validation; Investigation; Visualisation; Methodology; Writing—review and editing. **Sabrina Sofia Burgener**: Formal analysis; Validation; Investigation; Visualisation; Methodology; Project administration; Writing—review and editing. **Jakub Began**: Validation; Investigation; Methodology; Writing—review and editing. **Melan Kurera**: Validation; Investigation; Writing—review and editing. **Jing Rong Chia**: Investigation. **Emmanuelle Frampton**: Investigation. **Joanna Crawford**: Investigation; Writing—review and editing. **Monalisa Oliveira**: Investigation. **Kirsten M Kenney**: Investigation. **Jared R Coombs**: Investigation; Writing—review and editing. **Masahiro Yamamoto**: Resources. **Si Ming Man**: Resources; Supervision; Funding acquisition; Writing—review and editing. **Petr Broz**: Resources; Supervision; Funding acquisition; Writing—review and editing. **Pablo Pelegrin**: Resources; Supervision; Funding acquisition; Methodology; Project administration; Writing—review and editing. **Kate Schroder**: Resources; Supervision; Funding acquisition; Project administration; Writing—review and editing.

Source data underlying figure panels in this paper may have individual authorship assigned. Where available, figure panel/source data authorship is listed in the following database record: biostudies:S-SCDT-10_1038-S44318-025-00507-z.

## Disclosure and competing interests statement

As co-founder of Viva in vitro diagnostics, PP declares that the research presented herein was conducted in the absence of any commercial or financial relationships that could be construed as a potential conflict of interest. KS is a co-inventor on patent applications for NLRP3 inhibitors licensed to Inflazome Ltd., a company headquartered in Dublin, Ireland. Inflazome is developing drugs that target the NLRP3 inflammasome to address unmet clinical needs in inflammatory disease. KS served on the Scientific Advisory Board of Inflazome in 2016–2017, and serves as a consultant to Quench Bio, USA, and Novartis, Switzerland. The authors declare no additional competing interests.

# Expanded View Figures

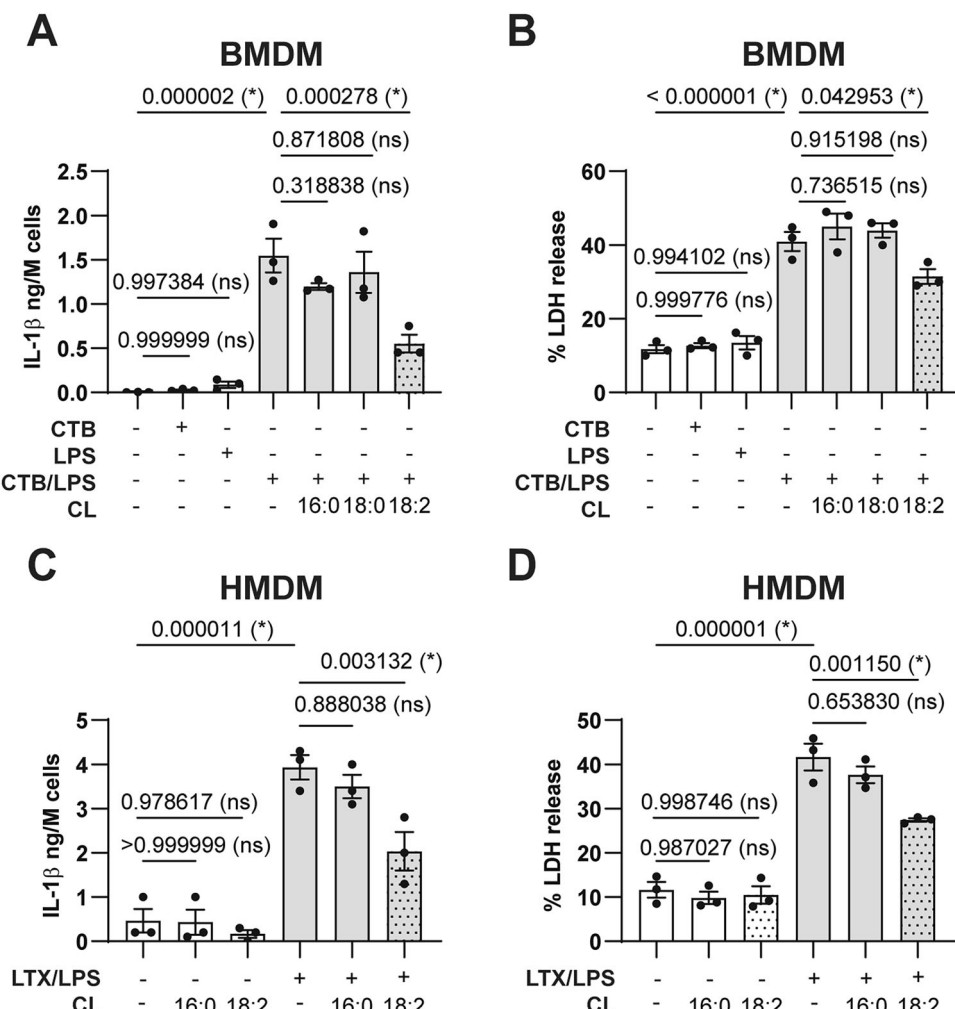

**Figure EV1. Unsaturated 18:2 but not saturated 16:0 and 18:0 CL inhibits noncanonical inflammasome signalling.**

(A, B) WT BMDM were incubated for 4 h with 1 μg/mL Pam₃CSK₄. Cell culture medium was then replaced with OptiMEM plus 20 μg/mL CTB, 2 μg/mL LPS, or 20 μg/mL CTB complexed with 2 μg/mL LPS (CTB/LPS), in the absence or presence of 10 μM saturated 16:0 CL, saturated 18:0 CL or unsaturated 18:2 CL. Cells were incubated for 18 h. Cleaved IL-1β was quantified in cell supernatants by ELISA (A). LDH release was quantified by cytotoxicity assay (B). (C, D) HMDM were incubated for 4 h with 0.1 μg/mL LPS. Cell culture medium was then replaced with OptiMEM or 20 μg/mL LTX complexed with 2 μg/mL LPS (LTX/LPS) in OptiMEM, in the absence or presence of 10 μM saturated 16:0 CL or unsaturated 18:2 CL. Cells were incubated for 4 h. Cleaved IL-1β was quantified in cell supernatants by ELISA (C). LDH release was quantified by cytotoxicity assay (D). Data information: Each symbol is the mean of three technical replicates from an independent biological replicate. Bars are the mean of three independent biological replicates (*n* = 3) ± SEM. Statistical analysis: Data were verified for normality using a Shapiro–Wilk test and analysed by one-way ANOVA Šídák's multiple comparisons test. *P* values are reported above bars. Statistical significance was defined as follows: significant difference for *P* < 0.05 (*), not significant for *P* ≥ 0.05 (ns).

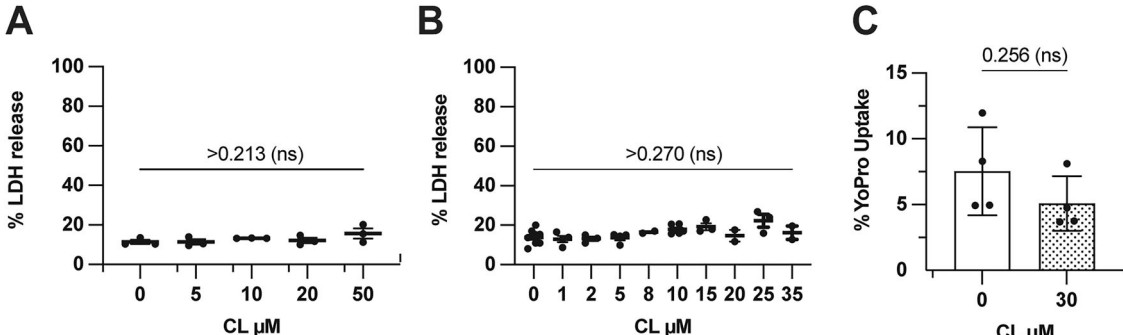

**Figure EV2. CL does not induce cell death.**

HMDM from healthy donors (**A**) or BMDM from wild-type mice (**B**, **C**) were incubated for 4 h with 1 µg/mL Pam₃CSK₄. Cell culture medium was then replaced with OptiMEM (0) or increasing CL concentrations (1 to 50 µM), and cells were incubated for 18 h. Lytic cell death was evaluated by LDH activity in supernatants (**A**, **B**) or YoPro uptake by lytic cells (**C**) and reported here as the percentage of cell lysis induced by 0.1% Triton. Data information: Each symbol is the mean of three technical replicates from an independent biological replicate. Bars are the mean of two or more independent biological replicates ($n = 2$ to 7) ± SEM. Statistical analysis: Data were verified for normality using a Shapiro–Wilk test and analysed by (**A**) one-way ANOVA, (**B**) Kruskal–Wallis test compared to control (0) ($n = 2$ not included in the statistical analysis), (**C**) unpaired *t* test. *P* values are reported above bars. Statistical significance was defined as follows: significant difference for $P < 0.05$ (*), not significant for $P \geq 0.05$ (ns).

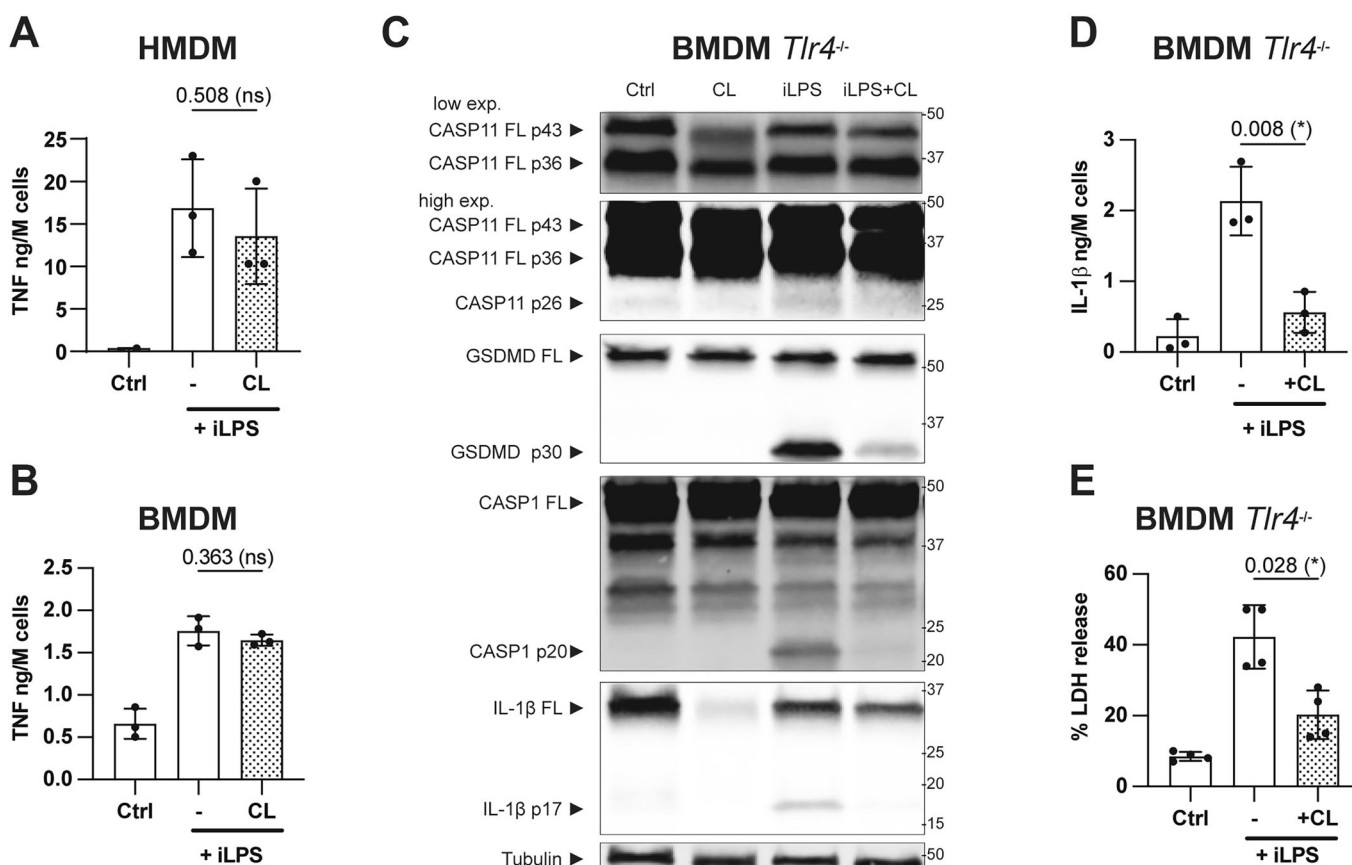

**Figure EV3. CL inhibits CASP11 activation independently of TLR4.**

HMDM from healthy donors (**A**), and BMDM from wild-type (**B**) or *Tlr4*$^{-/-}$ (**C–E**) mice were incubated for 4 h with 1 µg/mL Pam$_3$CSK$_4$. Cell culture medium was then replaced with OptiMEM or the noncanonical inflammasome activator LPS complexed with 0.25 % v/v LTX (**A**) or 20 µg/mL CTB (**B–E**) (iLPS) in the presence of HEPES (−) or 10 µM CL. Released TNF and cleaved IL-1β were measured in supernatants by ELISA (**A, B, D**). CASP11, GSDMD, IL-1β cleavage, and tubulin expression were assessed by western blot (**C**). LDH release was quantified by cytotoxicity assay (**E**). Data information: The blot is representative of three independent biological replicates (*n* = 3). Each symbol is the mean of three technical replicates from an independent biological replicate. Bars are the mean of three or more independent biological replicates (*n* = 3 to 4) ± SEM. Statistical analysis: Data were verified for normality using a Shapiro–Wilk test and analysed by (**A, E**) Mann–Whitney test, (**B, D**) unpaired *t* test. *P* values are reported above bars. Statistical significance was defined as follows: significant difference for *P* < 0.05 (*), not significant for *P* ≥ 0.05 (ns).

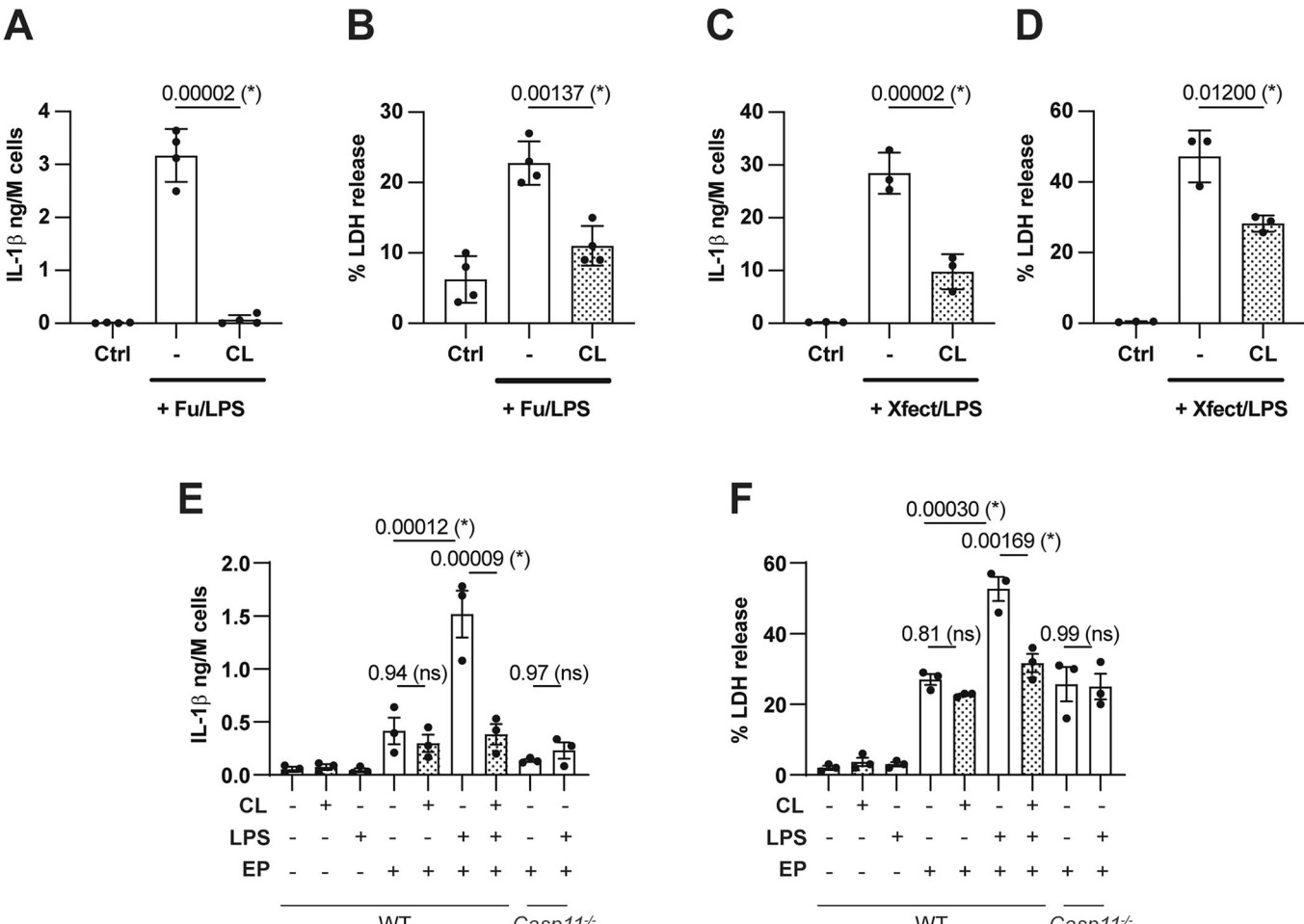

**Figure EV4. CL inhibits the noncanonical inflammasome regardless of the delivery system used for LPS.**

(A–D) BMDM from WT mice were incubated for 4 h with 1 µg/mL Pam₃CSK₄. Cell culture medium was then replaced with OptiMEM, or 2 µg/mL of the noncanonical inflammasome activator LPS from *Escherichia coli* B4 strain (EC-B4) complexed with 0.5% FuGENE HD (**A, B**) or 1.2% Xfect (**C, D**) (iLPS) in the presence of HEPES (−) or 10 µM CL. Cells were incubated for 18 h (**A, B**) or 4 h (**C, D**). Cleaved IL-1β was quantified in cell supernatants by ELISA. LDH release was quantified by cytotoxicity assay. (**E, F**) BMDM from wild-type or *Casp11*⁻/⁻ mice were incubated for 4 h with 1 µg/mL Pam₃CSK₄. Cell culture medium was then replaced with OptiMEM in the presence of HEPES (−), 10 µM CL, or 2 µg/mL of LPS. Cells were left untouched or electroporated ( + EP), then fresh OptiMEM was added, and cells were incubated for 4 h. Cleaved IL-1β was quantified in cell supernatants by ELISA. LDH release was quantified by cytotoxicity assay. Data information: Each symbol is the mean of technical triplicates from an independent biological replicate. Bars are the mean of three or more independent biological replicates (*n* = 3 to 4) ± SEM. Statistical analysis: Data were verified for normality using a Shapiro–Wilk test and analysed by (**A–D**) unpaired *t* test, (**E, F**) one-way ANOVA Šídák's multiple comparisons test. *P* values are reported above bars. Statistical significance was defined as follows: significant difference for *P* < 0.05 (*), not significant for *P* ≥ 0.05 (ns).

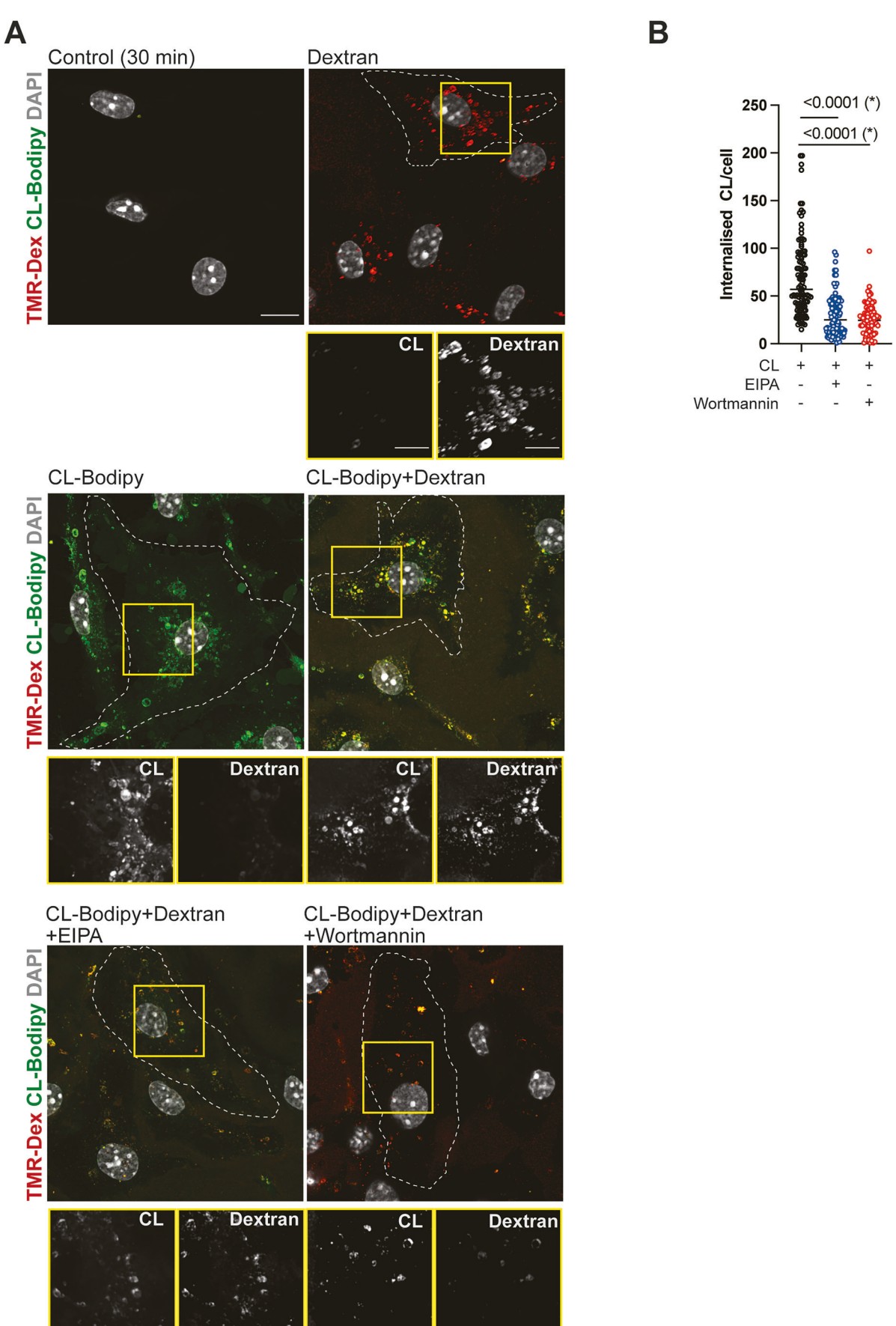

◀ **Figure EV5. CL accesses the cell interior by endocytosis to inhibit CASP11 activation in BMDM.**

(A) BMDM were primed for 4 h with 1 µg/mL $Pam_3CSK_4$ and incubated for 1 h in OptiMEM with EIPA (10 µM) or Wortmannin (10 µM) prior to the addition of HEPES (Control), 10 µM of CL liposomes containing 1% (w/w) TopFluor CL (CL-BODIPY), or 100 µg/mL 70 kDa TMR-dextran for 30 min. Macrophages were immunostained with phalloidin (for quantification) and DAPI (grey). Images are fixed-Airyscan confocal imaging of DAPI (grey), TMR-dextran (red) and CL-BODIPY (green). Cell outlines were drawn from brightfield. Data information: Images are maximum intensity projections of Z-stacks acquisitions and representative of three independent experiments ($n = 3$). Scale bar = 10 µm and 5 µm in inset panels. (B) The graph is the quantification of CL uptake: for quantification, images were segmented based on phalloidin staining, and internalised TopFluor CL was quantified in Fiji. Each dot represents the number of BODIPY-positive vesicles per cell from three independent biological replicates ($n = 3$). Statistical analysis: Data were verified for normality using a Shapiro–Wilk test and analysed by Mann–Whitney test. P values are reported above bars. Statistical significance was defined as follows: significant difference for $P < 0.05$ (*), not significant for $P \geq 0.05$ (ns).

