## [Peer Review File · The EMBO Journal]

Cardiolipin inhibits the non-canonical inflammasome by preventing LPS binding to caspase-4/11

Malvina Pizzuto, Mercedes Monteleone, Sabrina Burgener, Jakub Began, Melan Kurera, Jing Chia, Emmanuelle Frampton, Joanna Crawford, Monalisa Oliveira, Kirsten Kenney, Jared Coombs, Masahiro Yamamoto, Si Ming Man, Petr Broz, Pablo Pelegriin, and Kate Schroder

Corresponding author(s): Malvina Pizzuto (m.pizzuto@uq.edu.au)

Review Timeline:

Submission Date:	8th Jan 25
Editorial Decision:	21st Feb 25
Revision Received:	19th May 25
Editorial Decision:	5th Jun 25
Revision Received:	12th Jun 25
Accepted:	16th Jun 25

Editor: Ioannis Papaioannou

Transaction Report:

Dear Dr. Pizzuto,

Thank you for submitting your manuscript EMBOJ-2025-120114 for consideration by The EMBO Journal, and for your patience during peer review. Your manuscript has now been seen by two experts in the field, and we have received their comments, which you can find below.

As you will see, both referees indicate interest in the findings, and recognize that the study is well-developed and convincingly proves the capacity of cardiolipin to specifically interact with Casp4/11. However, they also identify a number of limitations regarding both the depth and clarity of the biochemical investigation and the extent to which the in vivo relevance of the results is supported by the available data. We agree with the referees that most of their suggestions would significantly improve the study and the manuscript and increase the impact of the work on the field.

Given the referees' positive comments and recommendations, I would like to invite you to revise your study taking the referees' suggestions on board. I would encourage you to send me early in the process of your revision a draft point-by-point response to the reviewers' comments explaining which of their criticisms (and how) you are willing and able to address during your revision. If there are any points you do not agree with or you cannot address, please let us know. Such a point-by-point response would be very helpful in discussing further with you a realistic and constructive revision plan. I should add that it is The EMBO Journal policy to allow only a single round of major revision, and acceptance of your manuscript will therefore depend on the completeness of your responses to the referees' comments and an appropriate revision plan.

When you are ready to submit your revised manuscript for re-review, please include a detailed point-by-point response addressing all referees' comments. We generally allow three months as standard revision time (May 20, 2025). As a matter of policy, competing manuscripts published during this period will not negatively impact our assessment of the conceptual advance presented by your study. However, we request that you contact us as soon as possible upon publication of any related work, to discuss how to proceed. Should you foresee a problem in meeting this three-month deadline, please let us know in advance and we may be able to grant an extension.

Thank you for the opportunity to consider your work for publication in The EMBO Journal. I look forward to your revision.

Best regards,

Ioannis

Instructions for preparing your revised manuscript

1. When you are ready to submit the revision, please upload:

- A Word file of the manuscript text (including legends of main Figures, EV Figures and Tables). Please make sure that changes are highlighted (or "tracked") to be clearly visible.

- Individual production-quality figure files (one file per figure). When assembling your figures, please refer to our figure preparation guidelines in order to ensure proper formatting and readability in print as well as on screen:

If the data shown in a figure are obtained from n {less than or equal to} 2, please use scatter plots showing the individual data points.

i. the name of the statistical test used to generate error bars and P values

ii. the number (n) of independent experiments (please specify technical or biological replicates) underlying each data point

(discussion of statistical methodology can be reported in the Materials and Methods section, but figure legends should contain a basic description of n , P , and the test applied)

iii. the nature of the bars and error bars (s.d., s.e.m.).

- A point-by-point response to the referees' comments, with a detailed description of the changes made (as a word file). All referees' concerns must be fully addressed and their suggestions taken on board. When preparing your letter of response to the referees' comments, please bear in mind that this will form part of the Review Process File and will therefore be available online to the community. Please note that you have the possibility to opt out of the transparent process at any stage prior to publication by letting the editorial office know (contact@embojournal.org); if you do opt out, the Review Process File link will point to the following statement: "No Review Process File is available with this article, as the authors have chosen not to make the review process public in this case.". For more details on our Transparent Editorial Process, please visit our website: <https://www.embopress.org/page/journal/14602075/authorguide#transparentprocess>

- Expanded View (EV) files (replacing Supplementary Information) that are collapsible/expandable online. A maximum of 5 EV Figures can be typeset. EV Figures should be cited as "Figure EV1, Figure EV2" etc. in the text, and their respective legends should be included in the manuscript file after the legends of regular figures. See detailed instructions regarding Expanded View files here: <https://www.embopress.org/page/journal/14602075/authorguide#expandedview>

- For the figures that you do NOT wish to display as Expanded View figures, they should be bundled together with their legends in a single PDF file called "Appendix", which should start with a short Table of Contents (including page numbers). Appendix figures should be referred to in the main text as: "Appendix Figure S1, Appendix Figure S2" etc. Please see detailed instructions here: <https://www.embopress.org/page/journal/14602075/authorguide#expandedview>

- A complete author checklist, which you can download from our author guidelines (<https://www.embopress.org/page/journal/14602075/authorguide>). Please note that the checklist will also be part of the Review Process File.

2. Please note that no statistics should be calculated and shown in Figures if $n=2$. Please also note that each p value should be reported as an exact value.

3. Before submitting your revision, primary datasets (and computer code, where appropriate) produced in this study need to be deposited in appropriate public databases (see <https://www.embopress.org/page/journal/14602075/authorguide#dataavailability>). The accession numbers, database, and the specific URLs (links) should be listed in a formal "Data availability" section (placed after Methods), following the example below:

"The RNA-seq datasets produced in this study are available in the following database:
Gene Expression Omnibus GSE46843 (<https://www.ncbi.nlm.nih.gov/geo/query/acc.cgi?acc=GSE46843>)"

*** All links should resolve to a page where the data can be accessed. ***

*** Please remember to provide in the Data availability section of your revised manuscript reviewer passwords if the datasets are not yet public. ***

*** The Data Availability Section is restricted to new primary data that are part of this study. In case you have no data that require deposition in a public database, please state so instead of referring to the database: "Our study includes no data deposited in public repositories." under the heading "Data availability". ***

4. The materials and methods need to be described in the manuscript using our structured methods format, which is now required for all research articles. According to this format, the Methods section includes a single "Reagents and Tools Table" - listing key reagents, experimental models, software and relevant equipment including their sources and relevant identifiers- followed by a "Methods and Protocols" section describing the methods. Please download and fill our Reagents and Tools Table template (.docx), which you can find in our author guide:

<https://www.embopress.org/page/journal/14602075/authorguide#structuredmethods>. When submitting your revised manuscript, please do not include the Reagents and Tools Table in the Methods section of the manuscript but instead upload it as a separate file choosing the file type "Reagent Table".

5. Please check that the title and the abstract of the manuscript are brief, yet explicit, even to non-specialists. The length of the title should not exceed 100 characters, and the abstract should be a single paragraph not exceeding 175 words.

6. Please also note our reference format: <https://www.embopress.org/page/journal/14602075/authorguide#referencesformat>.

8. Please remember: digital image enhancement is acceptable practice, as long as it accurately represents the original data and conforms to community standards. If a figure has been subjected to significant electronic manipulation, this must be noted in the

figure legend or in the "Materials and Methods" section. The editors reserve the right to request original versions of figures and the original images that were used to assemble the figure.

9. Our journal encourages inclusion of data citations in the reference list to directly cite datasets that were obtained from public databases. Data citations in the article text are distinct from normal bibliographical citations and should directly link to the database records from which the data can be accessed. In the main text, data citations are formatted as follows: "Data ref: Smith et al, 2001" or "Data ref: NCBI Sequence Read Archive PRJNA342805, 2017". In the Reference list, data citations must be labeled with "[DATASET]". A data reference must provide the database name, accession number/identifiers, and a resolvable link to the landing page from which the data can be accessed at the end of the reference. Further instructions are available at: <https://www.embopress.org/page/journal/14602075/authorguide#referencesformat>.

10. We request authors to consider both actual and perceived competing interests. Please review our policy (<https://www.embopress.org/page/journal/14602075/authorguide#conflictofinterest>) and update your competing interests statement if necessary. Please name this section 'Disclosure and competing interests statement' and place it after the Acknowledgements section.

11. Please note that all corresponding authors are required to provide an ORCID ID upon submission of a revised manuscript (<https://orcid.org/>). Please find instructions on how to link your ORCID ID to your account in our manuscript tracking system in our Author guidelines (<https://www.embopress.org/page/journal/14602075/authorguide#authorshipguidelines>).

12. We use CRediT to specify the contributions of each author in the journal submission system. CRediT replaces the author contribution section, which should be removed from the manuscript. Please use the free text box to provide more detailed descriptions. See also guide to authors: <https://www.embopress.org/page/journal/14602075/authorguide#authorshipguidelines>.

14. We would also welcome the submission of cover suggestions or motifs to be used by our Graphics Illustrator in designing a cover.

15. Please use the link below to submit your revision:
<https://emboj.msubmit.net/cgi-bin/main.plex>

Best regards,

Ioannis

Referee #1:

In this study, the authors report a biochemical interaction between cardiolipin (CL) and murine caspase-11 (the homologue of caspase-4/5 in humans). This is an interesting study, with detailed mechanistic analyses performed. Several suggestions are offered below, which would need to be addressed in order to validate the central findings of this work.

1. In the author's prior work, they showed that CL with saturated acyl chains is a TLR4 agonists, whereas CL with unsaturated acyl chains is a TLR4 antagonist. As the authors use their prior work on TLR4 to justify this current study on caspase-4, the logical question arises of whether CL saturation impacts caspase-4 function. The authors are encouraged to perform these studies.

2. Similar to point 1, the authors state on page 4 that "CL occupies the LPS binding site of TLR4". Reference 42, was referred to in this statement. However, no TLR4 binding assays were performed in reference 42. Prior work suggested that LPS antagonists bind CD14 without interacting with TLR4, leading to the endocytosis of CD14 and a subsequent lack of TLR4 signaling capacity (PMID: 26546281). Does saturated or unsaturated CL also promote CD14 endocytosis without binding TLR4?

3. The experiment presented in Figure 2E is confusing. The authors show in Figure 2D that CL prevents LPS interactions with

caspase-11. Since the outcome of LPS interactions is dimerization of caspase-11, CL would be expected to block caspase-11 dimer formation. In Figure 2E, the authors used a caspase-11 variant that can be dimerized with a chemical AP20187. This chemical inducible dimerization would be expected to bypass the ability of CL to prevent caspase-11 dimerization. However, the data presented indicates that CL blocks AP20187 induced caspase activation. This finding raises questions of the precise mechanism of CL action. The authors are encouraged to explain this confusing finding.

4. Biochemical analysis of caspase-4/5 interactions with CL should be performed, as much of the work is in human and mouse systems. Yet only caspase-11 was studied biochemically.

Referee #2:

The work from Pizzuto and colleagues describes the capacity of cardiolipin (CL) to interact with the CARD domain of Casp4/11, preventing the binding of LPS, and thus acting as a non-canonical, rather than global, inflammasome inhibitor.

This is a very elegant study that makes use of multiple techniques and that convincingly proves the capacity of CL to specifically interact with Casp4/11. Beside several minor points reported below, my major suggestion would be to implement a slightly different in vivo model which can be utilized even in the Tlr4^{-/-} background, making the in vivo relevance of the findings even more compelling.

-The increased release of IL-1 and LDH by RS-LPS, especially in BMDM, is puzzling. Is there any effect on NFκB activation and/or other cytokine production? If not antagonistic, once would expect it to be at least neutral.

-Loss of Casp4 in Fig.2A is not evident, can the authors quantify the protein over multiple replicates? Similarly, it is not clear in Fig. 2B whether Casp11 is directly affected or not by CL treatment.

-Authors nicely demonstrate CL entrance in the cell. How does (extracellularly delivered) CL enter the cell to interact with cytoplasmic Casp4/11 thus preventing transfected LPS activity? Is it an active transport or a passive process? If not experimentally, this aspect should be at least discussed.

-Authors convincingly show CL localizes into mitochondria. CL in mitochondria has been previously shown to activate Casp8. Is Casp8 involved in any of the processes described by the authors?

-Experiment shown in Fig. S3H-J demonstrate that iLPS does not require GBPs to activate Casp11/1. It is not clear how this experiment excludes that CL might also interact with GBPs. To assess this possibility, the experiment should be repeated with bacteria known to be sensitive to the activity of GBPs to activate Casp11 via their LPS. If this is not of interest for the authors, I'd rather not include the data shown in figure S3H-J.

-Although the authors looked at TNF in their in vivo experiment (Fig. 4), the use of LPS to prime and activate Casp11/1 in the presence of CL that can also, at least to some extent, prevent TLR4 activation poses some concerns. Other inflammatory mediators, such as IL6 and IFNs, should be measured. Also, the authors should perform the in vivo experiment with the prime with poly:IC, followed by a low dose of LPS, possibly in Tlr4^{-/-} mice (as originally done to identify Casp11 as a receptor for LPS).

-The last paragraph describing Fig. 4 is probably better suited for being part of the "Conclusions and Perspectives".

Dear Editor and Dear Referees,

Thank you for giving us the possibility to revise our manuscript EMBOJ-2025-120114.

We have reworked all formatting to conform to the EMBO journal style, changed our spelling to British English, and addressed all referees' points.

Below in **blue** is a point-by-point response to the referees' comments explaining how we have addressed their questions in the manuscript revision and with new experiments.

Main changes are reported below and highlighted in blue font in the manuscript text.

Thank you again for your valuable feedback and consideration.

Referee #1:

In this study, the authors report a biochemical interaction between cardiolipin (CL) and murine caspase-11 (the homologue of caspase-4/5 in humans). This is an interesting study, with detailed mechanistic analyses performed. Several suggestions are offered below, which would need to be addressed in order to validate the central findings of this work.

We thank the referee for their positive comments and support for our study, and address specific comments below.

1. In the author's prior work, they showed that CL with saturated acyl chains is a TLR4 agonist, whereas CL with unsaturated acyl chains is a TLR4 antagonist. As the authors use their prior work on TLR4 to justify this current study on caspase-4, the logical question arises of whether CL saturation impacts caspase-4 function. The authors are encouraged to perform these studies.

Our revised manuscript provides new data showing that, unlike unsaturated CL (18:2), which suppresses CASP4/11 function, saturated forms of CL (16:0 and 18:0) do not reduce CASP4/11 signalling outputs (LDH and IL-1 β release) in murine macrophages (BMDM) and human macrophages (HMDM) (Rebuttal Figure R1/Expanded Figure EV1). This indicates that indeed, CL saturation degree is critical for its CASP4/11 inhibitory function, with unsaturated CL suppressing CASP4/11 while saturated CL does not. The first paragraph of the results section has been modified accordingly.

Rebuttal Figure R1/Expanded Figure EV1: Unsaturated 18:2 but not saturated 16:0 and 18:0 CL inhibit noncanonical inflammasome signalling

A-B WT BMDM were incubated for 4 hours with 1 μ g/mL Pam₃CSK₄. Cell culture medium was then replaced with OptiMEM plus 20 μ g/mL CTB, 2 μ g/mL LPS, or 20 μ g/mL CTB complexed with 2 μ g/mL LPS (CTB/LPS), in the absence or presence of 10 μ M saturated 16:0 CL, saturated 18:0 CL or unsaturated 18:2 CL. Cells were incubated for 18 hours. Cleaved IL-1 β was quantified in cell supernatants by ELISA (A). LDH release was quantified by cytotoxicity assay (B). **C-D** HMDM were incubated for 4 hours with 0.1 μ g/mL LPS. Cell culture medium was then replaced with OptiMEM or 20 μ g/mL LTX complexed with 2 μ g/mL LPS (LTX/LPS) in OptiMEM, in the absence or presence of 10 μ M saturated 16:0 CL or unsaturated 18:2 CL. Cells were incubated for 4 hours. Cleaved IL-1 β was quantified in cell supernatants by ELISA (C). LDH release was quantified by cytotoxicity assay (D).

Data information: Each symbol is the mean of three technical replicates from an independent biological replicate. Bars are the mean of three independent biological replicates (n = 3) \pm SEM.

Statistical analysis: Data were verified for normality using a Shapiro-Wilk test and analysed by one-way ANOVA Šídák's multiple comparisons test. P-values are reported above bars. Statistical significance was defined as follows: significant difference for p<0.05 (*), not significant for p \geq 0.05 (ns).

2. Similar to point 1, the authors state on page 4 that "CL occupies the LPS binding site of TLR4". Reference 42, was referred to in this statement. However, no TLR4 binding assays were performed in reference 42. Prior work suggested that LPS antagonists bind CD14 without interacting with TLR4, leading to the endocytosis of CD14 and a subsequent lack of TLR4 signaling capacity (PMID: 26546281). Does saturated or unsaturated CL also promote CD14 endocytosis without binding TLR4?

We did not assess CD14 or TLR4 endocytosis in our previous study. Our assertion that CL occupies the LPS binding site of TLR4 is based on the competition assay described in Ref. 42 (Pizzuto et al., 2019). In this assay, increasing LPS concentrations in the presence of a fixed CL concentration restored NF- κ B signaling to the same plateau (LPS = LPS + CL). This profile is not consistent with a mechanism of action in which CL promotes CD14 endocytosis to block TLR4 signalling capacity. Rather, this profile suggests a competitive antagonist mechanism, where CL is displaced by LPS. Furthermore, molecular docking analyses in Ref. 42 indicated that both saturated and unsaturated CL can occupy the LPS binding site, with unsaturated CL adopting a conformation similar to the TLR4 antagonist eritoran, while saturated CL resembles the LPS-bound agonist conformation.

The scope of the current study is to investigate CL regulation of CASP4/11, not TLR4. However, we acknowledge the referee's point that CL binding to TLR4 was not formally demonstrated in the referenced study, so we have revised the text to provide greater clarity.

Please note that Ref 42 is now (Pizzuto et al., 2019) and that we have added the suggested reference PMID: 26546281 as (Tan et al., 2015).

Original submission text: *We showed that CL occupies the LPS binding site of TLR4 and specifically inhibits TLR4 without affecting signaling by other TLRs⁴²*

Revised text: *We showed that unsaturated CLs specifically inhibit TLR4 signalling without affecting signalling by other TLRs, while saturated CLs activate TLR4 (Pizzuto et al., 2019). Competition tests and molecular docking suggested that CL likely occupies the LPS binding site of TLR4 (Pizzuto et al., 2019). If this mechanism-of-action is confirmed by binding studies, it would rule out other possible activities, such as promoting CD14 endocytosis (Tan et al., 2015).*

3. The experiment presented in Figure 2E is confusing. The authors show in Figure 2D that CL prevents LPS interactions with caspase-11. Since the outcome of LPS interactions is dimerization of caspase-11, CL would be expected to block caspase-11 dimer formation. In Figure 2E, the authors used a caspase-11 variant that can be dimerized with a chemical AP20187. This chemical inducible dimerization would be expected to bypass the ability of CL to prevent caspase-11 dimerization. However, the data presented indicates that CL blocks AP20187 induced caspase activation. This finding raises questions of the precise mechanism of CL action. The authors are encouraged to explain this confusing finding.

We indeed showed that CL blocks LPS interaction with CASP11 (Fig 2D) and prevents AP20187-induced caspase activation (Fig 2E), and apologise if our discussion on this point was unclear. Our revised manuscript clarifies our proposed mechanism of CL action, based on our observations that CL binds the CARD domain (Fig 2G) and that the CARD is required for CL to suppress AP-induced caspase activation (Fig 2E vs 2F):

"CL binds to the CARD, and we propose this binding thereby (i) blocks LPS from binding to the CARD, and thus inhibits LPS-induced CASP4/11 activation, and (ii) sterically impedes artificial dimerisation using the AP drug - either by blocking the capacity of the AP drug to simultaneously bind to two DmrB

domains, or by sterically impeding the AP/DmrB-induced dimerisation of the CARD and/or the protease domain.”

4. Biochemical analysis of caspase-4/5 interactions with CL should be performed, as much of the work is in human and mouse systems. Yet only caspase-11 was studied biochemically.

We apologise for the confusion and would like to clarify that most of our biochemical characterization in our original submission was performed using CASP4, not CASP11. Specifically, the constructs used in Figs. 2E, F, and G are CASP4, not CASP11. To provide the reader with greater clarity, we have added titles above data panels in the figure (Figure 2).

Referee #2:

The work from Pizzuto and colleagues describes the capacity of cardiolipin (CL) to interact with the CARD domain of Casp4/11, preventing the binding of LPS, and thus acting as a non-canonical, rather than global, inflammasome inhibitor.

This is a very elegant study that makes use of multiple techniques and that convincingly proves the capacity of CL to specifically interact with Casp4/11. Beside several minor points reported below, my major suggestion would be to implement a slightly different in vivo model which can be utilized even in the Tlr4^{-/-} background, making the in vivo relevance of the findings even more compelling.

We thank the referee for their positive comments and support for our study, and address specific comments below.

-The increased release of IL-1 and LDH by RS-LPS, especially in BMDM, is puzzling. Is there any effect on NFκB activation and/or other cytokine production? If not antagonistic, one would expect it to be at least neutral.

Indeed, Fig 1A-B shows that RS-LPS did trend towards inducing IL-1β release in BMDM and promoted iLPS-induced LDH release in BMDM and HMDM. We agree that this RS-LPS data was unexpected.

While the potential effects of RS-LPS on TLR4 signalling are not the focus of our study, we were intrigued by the BMDM data and have performed several experiments to address this question.

Our data suggest that while RS-LPS antagonises human TLR4 signalling, RS-LPS may function as a partial agonist of murine TLR4; accordingly, in BMDM, RS-LPS weakly activates NF-κB in the absence of LPS but blocks TLR4 signalling in the presence of LPS (Rebuttal Figure R2/Appendix Figure S1). Fig 1B data is consistent with such a function for RS-LPS in Pam₃CSK₄-primed BMDM; here, we propose that RS-LPS weakly activates TLR4 to enhance cell priming, and boost IL-1β and LDH release induced by intracellular LPS-CASP11 signalling. While NF-κB activation provides a potential explanation for how RS-LPS boosts iLPS-induced CASP11 signalling, whether RS-LPS also activates CASP4/11 remains an open question for future studies. We do not address this here, as our study focuses on CASP4/11 inhibition.

We have incorporated the TNF data presented in Rebuttal Figure R2 into our revised manuscript as Appendix Figure S1 and added new text as follows:

“To investigate how RS-LPS affected TLR4-induced NF-κB activation, we treated BMDM and HMDM with RS-LPS without other stimuli for 4 hours and measured TNF secretion. RS-LPS induced TNF release

in BMDM but not in HMDM (Appendix Fig S1), suggesting that BMDM exposure to RS-LPS in the absence of other stimuli may trigger NF- κ B activation. This was unexpected because RS-LPS is described as a TLR4 antagonist (Rose et al., 1995). To further investigate this, we treated macrophages with RS-LPS in the presence or absence of LPS. In both BMDM and HMDM, RS-LPS inhibited TNF release induced by extracellular LPS (Appendix Fig S1), confirming the quality of our ultrapure RS-LPS preparation and its reported function as an antagonist of LPS-TLR4 signalling. These data collectively suggest that while RS-LPS is an antagonist of human TLR4, RS-LPS could be a partial agonist for murine TLR4. This is not unprecedented amongst lipid modulators of TLR4; for example, lipid IVa is also a partial agonist of murine TLR4 but an antagonist of human TLR4 (Walsh et al., 2008). Given that NF- κ B signalling licenses CASP11 for signalling, RS-LPS-induced NF κ B activity provides a potential explanation for how RS-LPS boosts iLPS-induced CASP11 signalling in BMDM. Whether RS-LPS also activates the noncanonical inflammasome remains an open question for future studies.”

Rebuttal Figure R2/Appendix Figure S1: RS-LPS induces TNF secretion in BMDM but not in HMDM, and inhibits LPS-induced TNF secretion in both HMDM and BMDM

HMDM (A) or BMDM (B) were incubated for 4 hours with 0.1 μ g/mL LPS in the absence or presence of 2 μ g/mL RS-LPS. TNF was quantified in cell supernatants by ELISA.

Data information: Each symbol is the mean of three technical replicates from an independent biological replicate. Bars are the mean of three independent biological replicates (n = 3) \pm SEM.

Statistical analysis: Data were verified for normality using a Shapiro-Wilk test and analysed by one-way ANOVA Šídák's multiple comparisons test. P-values are reported in the figure. Significant difference for $p < 0.05$ (*), not significant $p \geq 0.05$ (ns).

-Loss of Casp4 in Fig.2A is not evident, can the authors quantify the protein over multiple replicates? Similarly, it is not clear in Fig. 2B whether Casp11 is directly affected or not by CL treatment.

We quantified CASP4 and CASP11 band intensities across three replicates and presented the data as fold increase relative to untreated cells (Appendix Figures S4 and S5).

Although statistical significance was not always reached, trends were robust across biological replicate experiments, and support our conclusions.

Specifically, full-length (FL) CASP4 levels decrease following intracellular LPS stimulation, and this reduction is reversed by CL treatment (Rebuttal Figure R3 / Appendix Figure S4).

Similarly, the trend in FL-CASP11 levels supports our conclusion that Pam₃CSK₄-priming induces CASP11 expression, which is not altered by CL (Rebuttal Figure R4A / Appendix Figure S5A).

Importantly, quantification of cleaved CASP11 reveals that intracellular LPS significantly increases CASP11 cleavage, and this effect is significantly inhibited by CL (Rebuttal Figure R4 B / Appendix Figure S5 B).

Rebuttal Figure R3/Appendix Figure S4: CL does not affect basal full-length CASP4 expression while restoring iLPS-induced decrease - Western Blot Quantification related to Figure 2

HMDM were incubated for 4 hours with cell culture medium (unprimed) or 1 µg/mL Pam₃CSK₄ (all other conditions). Cell culture medium was then replaced with OptiMEM, LTX 0.25% v/v, 10 µM CL, or 2 µg/mL of LPS complexed with 0.25% v/v LTX in the presence of HEPES vehicle or 10 µM CL. Cells were incubated for 4 hours. The amount of full-length CASP4 was assessed in mixed supernatants and lysates by western blot. Band intensity was quantified using ImageLab (BioRad) (adjusted volumes) and reported here as fold increase compared to untreated cells.

Data information: Each symbol is the measure of an independent biological replicate. Bars are the mean of three independent biological replicates (n = 3) ± SEM.

Statistical analysis: Data were verified for normality using a Shapiro-Wilk test and analysed by repeated measures (RM) one-way ANOVA (paired) Šídák's multiple comparisons test. P-values are reported in the figure. Significant difference for p<0.05 (*), not significant p≥0.05 (ns).

Rebuttal Figure R4/Appendix Figure S5: CL decreases iLPS-induced cleaved but not full-length CASP11- Western Blot Quantification related to Figure 2

BMDM were incubated for 4 hours with cell culture medium (unprimed) or 1 µg/mL Pam₃CSK₄ (all other conditions). Cell culture medium was then replaced with OptiMEM, FuGENE HD 0.5% v/v, 10 µM CL, or 2 µg/mL of LPS alone or complexed with 0.5% FuGENE HD in the presence of HEPES vehicle or 10 µM CL. Cells were incubated for 4 hours. The amount of full-length (FL) (A) and cleaved (cl) (B) CASP11 was assessed in mixed supernatants and lysates by western blot. Band intensity was quantified using ImageLab (BioRad) (adjusted volumes) and reported here as fold increase compared to untreated cells.

Data information: Each symbol is the measure of an independent biological replicate. Bars are the mean of three independent biological replicates (n = 3) ± SEM.

Statistical analysis: Data were verified for normality using a Shapiro-Wilk test and analysed by repeated measures (RM) one-way ANOVA (paired) Šidák's multiple comparisons test. P-values are reported in the figure. Significant difference for p<0.05 (*), not significant p≥0.05 (ns).

-Authors nicely demonstrate CL entrance in the cell. How does (extracellularly delivered) CL enter the cell to interact with cytoplasmic Casp4/11 thus preventing transfected LPS activity? Is it an active transport or a passive process? If not experimentally, this aspect should be at least discussed.

This is an excellent question, and one that had also piqued our curiosity.

Lipid internalisation depends on cell type and liposome properties such as size, charge, and membrane fluidity, and can occur via endocytosis, phagocytosis, membrane fusion, or protein-mediated transport (Nel et al., Nat Mater 2009; Gandek et al., Adv Healthc Mater 2023).

LPS-containing vesicles are internalised by macrophages through endocytosis and subsequently escape from early endosomes to engage CASP11 (Vanaja et al., Cell 2016; Kunsmann et al., Sci Rep

2015; Parker et al., Infect Immun 2010). We anticipated that CL liposomes would follow a similar uptake route in macrophages.

We conducted experiments to confirm that BMDM internalise CL liposomes via endocytosis. We monitored the uptake of fluorescently labelled CL (CL-Bodipy) alongside Dextran Blue at an early time point (30 minutes) and assessed the effect of endocytosis inhibitors EIPA and Wortmannin on CL uptake. The new data demonstrate that CL is actively internalised by macrophages through endocytic pathways (Rebuttal Figure R6/ Expanded Fig EV5).

The following figure and text have been added to the manuscript:

“Lipid internalisation depends on cell type and liposome properties (size, charge, fluidity) and occurs via endocytosis, which includes membrane fusion, or protein-mediated transport (Gandek et al., 2023; Nel et al., 2009; Salloum et al., 2023). LPS vesicles are taken up by macrophages via endocytosis and then escape early endosomes to interact with CASP11 (Kunsmann et al., 2015; Parker et al., 2010; Vanaja et al., 2016). To investigate whether CL vesicles follow a similar uptake route into macrophages, we used TMR-Dextran 70 kDa as a marker of endocytosis (Li et al., 2015). Co-incubation of cells with CL-bodipy with TMR-Dextran 70 kDa for 30 minutes showed that CL co-localises with Dextran within macrophages (Fig EV5). Moreover, pre-incubation of macrophages with the endocytosis inhibitors Amiloride (EIPA) or Wortmannin (Kjeken et al., 2001; Koivusalo et al., 2010) significantly reduced CL uptake by macrophages (Fig EV5). Collectively, these data demonstrate that extracellular CL is actively internalised by macrophages through endocytosis.

Endocytosis is an established route for liposomal delivery to organelles (Matthaeus & Taraska, 2021; Popescu et al., 2006; Popov, 2022). A previous study showed that exogenous CL localises to mitochondria using the mitochondrial dye MitoTracker (Ikon et al., 2015), which we confirmed by showing that 18 hours after exposure, CL-Bodipy co-localised with TOMM20 in macrophage mitochondria (Fig 2 C and Appendix Fig S7). Thus, CL is taken up by cells and accumulates in the mitochondria, presumably in the mitochondrial outer membrane from which it has access to cytosolic proteins such as CASP4/11.”

Rebuttal Figure R6/Expanded Figure EV5: CL accesses the cell interior by endocytosis to inhibit CASP11 activation in BMDM

BMDM were primed for 4 hours with 1 $\mu\text{g}/\text{mL}$ Pam₃CSK₄ and incubated for 1 hour in OptiMEM with EIPA (10 μM) or Wortmannin (10 μM) prior to the addition of HEPES (Control), 10 μM of CL liposomes containing 1% (w/w) TopFluor CL (CL-BODIPY), or 100 $\mu\text{g}/\text{mL}$ 70KDa TMR-dextran for 30 min. Macrophages were immunostained with phalloidin (B, for quantification) and DAPI (grey). Images are fixed-Airyscan confocal imaging of DAPI (grey), TMR-dextran (red), and CL-Bodipy (green).

Data information: Images are maximum intensity projections of Z-stacks acquisitions and representative of three independent experiments (n=3). Scale bar = 10 μm and 5 μm in inset panels. The graph is the quantification of CL uptake: for quantification, images were segmented based on phalloidin staining, and internalised TopFluor CL was quantified in Fiji. Each dot represents the number of BODIPY-positive vesicles per cell from 3 independent biological experiments (n=3).

After endocytosis at an early time point, after 18 h CL localises to the mitochondria (Fig 2C and S3G—now Appendix Fig S6). In the revised manuscript, we extend these findings to HMDM by presenting new data showing that extracellular CL is internalised and colocalises with mitochondria (Rebuttal Fig R5 / Appendix Fig S7).

Rebuttal Figure R5/Appendix Figure S7: CL is internalised and colocalises with mitochondria in human macrophages.

Fixed-AiryScan confocal imaging of HMDM primed for 4 hours with 1 $\mu\text{g}/\text{mL}$ Pam₃CSK₄ and incubated with HEPES (Control) or 10 μM of TopFluor CL (CL-BODIPY) for a further 18 hours. Macrophages were immunostained with TOMM20 (red), CL-BODIPY (green), and DAPI (grey).

Data information: Images are maximum intensity projections of Z-stack acquisitions. Scale bar = 10 μm . Scale bar inset = 5 μm . Images are representative of three independent experiments (n = 3).

-Authors convincingly show CL localizes into mitochondria. CL in mitochondria has been previously shown to activate Casp8. Is Casp8 involved in any of the processes described by the authors?

[Supporting data not shown here due to confidentiality. These were made available to the referees and editor for evaluation.]

-Experiment shown in Fig. S3H-J demonstrate that iLPS does not require GBPs to activate Casp11/1. It is not clear how this experiment excludes that CL might also interact with GBPs. To assess this possibility, the experiment should be repeated with bacteria known to be sensitive to the activity of GBPs to activate Casp11 via their LPS. If this is not of interest for the authors, I'd rather not include the data shown in figure S3H-J.

We apologise for the lack of clarity regarding the experiment in Fig. S3H-J (now Appendix Fig S8), and agree it remains possible that CL interacts with GBPs. Potential interactions between CL and GBPs are not directly relevant to our study, as our conclusion from Fig. S3H-J (now Appendix Fig S8) is that GBPs are dispensable for CL-mediated inhibition of iLPS. We have reworded the text to clarify this point as follows:

“These data do not rule out the possibility that CL interacts with GBPs, but demonstrate that these GBPs are dispensable for the suppressive effect of CL on noncanonical inflammasome signalling.”

-Although the authors looked at TNF in their *in vivo* experiment (Fig. 4), the use of LPS to prime and activate Casp11/1 in the presence of CL that can also, at least to some extent, prevent TLR4 activation poses some concerns. Other inflammatory mediators, such as IL6 and IFNs, should be measured.

We understand the concern that CL may block the priming rather than the triggering step of CASP11 signalling in our *in vivo* experiment, however, our earlier findings demonstrated that CL does not affect TLR4 activation induced by high LPS concentrations (Now Fig 4D, Fig EV3 A-B and a previous study (Pizzuto et al., 2019)).

To support that CL does not inhibit the TLR4 signalling *in vivo*, we have measured IFN- γ and IL-6 levels in sera from our *in vivo* experiment (now Fig 4 B-C/Rebuttal Fig R8), and provided further *in vitro* data with the LPS strain used in our *in vivo* experiment (Appendix Fig S9/Rebuttal Fig R9).

New text and figures are reported below:

*“LPS induced significant serum IFN- γ , IL-6, and TNF levels, which were not dependent on CASP11 and not suppressed by CL (Fig 4 B-D). These data are in line with our earlier findings that CL does not affect TLR4 activation induced by high LPS concentrations (Fig EV3 A-B and a previous study (Pizzuto et al., 2019)) and suggest that CL does not prevent TLR4-dependent priming *in vivo*. Further, we treated BMDM with increasing concentrations of the same LPS strain used in our *in vivo* model (LPS from *Pseudomonas aeruginosa*, PA-LPS), in the presence of CL (Appendix Fig S9). TLR4 is exquisitely sensitive to LPS; 10 ng/ml extracellular LPS induces sub-maximal signalling inhibited by 10 μ M CL, while 100 ng/ml LPS induces maximal signalling and this and higher doses are not inhibited by 10 μ M CL. By contrast, at 2 μ g/mL, PA-LPS induced CASP11-dependent IL-1 β release, which was significantly reduced by 10 μ M CL (Appendix Fig S9).”*

Rebuttal Figure R8/Figure 4: Cardiolipin mitigates endotoxemia-induced systemic IL-1 β and weight loss *in vivo*.

WT and *Casp11*^{-/-} mice were weighed and injected (*i.p.*) with HEPES or 25 μ g/g CL. 10 minutes later, mice were *i.p.* challenged with PBS or 10 μ g/g LPS. After 6 hours, the animal's weight was recorded again, and blood was collected. Body weight loss was calculated as the percentage of the initial weight (A). IFN γ (B), IL-6 (C), TNF (D), and IL-1 β (E) were quantified in sera by ELISA.

Data information: Violin plot of data from four different cohorts of mice (individual mice from each cohort shown as color-matched data points). WT PBS and WT PBS + CL n=3; WT HEPES + LPS n=13; WT CL + LPS n=13; and *Casp11*^{-/-} HEPES + LPS n = 17.

Statistical analysis: Data were verified for normality using a Shapiro-Wilk test and analysed by **A and C** Kruskal-Wallis with Dunn's multiple comparisons test, **B, D and E** one-way ANOVA Sidak's multiple comparisons test. P-values are reported above bars. Statistical significance was defined as follows: significant difference for p<0.05 (*), not significant for p \geq 0.05 (ns).

Rebuttal Figure R9/Appendix Figure S9: CL does not inhibit TNF secretion induced by 1 µg/mL PA-LPS, but does suppress CASP11-dependent IL-1β secretion.

A BMDMs were incubated for 4 hours with increasing concentration of PA-LPS in the absence or presence of 10 µM CL. TNF was quantified in cell supernatants by ELISA.

B WT or *Casp11*^{-/-} BMDMs were incubated for 4 hours with 1 µg/mL Pam3CSK4. Cell culture medium was then replaced with OptiMEM plus 0.5% Fugene complexed with 1 µg/mL PA-LPS (Fu/PA-LPS), in the absence or presence of 10 µM CL. Cells were incubated for 18 hours. Cleaved IL-1β was quantified in cell supernatants by ELISA.

Data information: Each symbol is the mean of three technical replicates from an independent biological replicate. Bars are the mean of three or more independent biological replicates (n = 3) ± SEM.

Statistical analysis: Data were verified for normality using a Shapiro-Wilk test and analysed by **A** two-way ANOVA Tukey's multiple comparisons test, and **B** one-way ANOVA Šidák's multiple comparisons test. P-values are reported in the figure. Significant difference for p < 0.05 (*), not significant p ≥ 0.05 (ns).

Also, the authors should perform the *in vivo* experiment with the prime with poly:IC, followed by a low dose of LPS, possibly in *Tlr4*^{-/-} mice (as originally done to identify *Casp11* as a receptor for LPS).

We agree that the referee's proposed *in vivo* experiment would provide definitive evidence to rule out a TLR4-inhibitory function for CL. However, we are confident that the additional data we have provided (the inclusion of other *in vivo* cytokines as suggested by the referee and further supporting *in vitro* experiments) adequately address this concern. Unfortunately, it was not feasible for us to establish an entirely new *in vivo* model within the 3-month revision period set by The EMBO Journal. Developing such a model would require significant delays due to the time needed to obtain ethics approval (typically 3–4 months at our institution) and the additional time required for mouse breeding and establishing this experimental protocol.

In the revised manuscript, we discuss the lack of *in vivo* validation as a limitation of the work:

“Limitations of the Study: We were unable to test the impact of CL in a TLR4-independent in vivo endotoxemia model (e.g., in *Tlr4*^{-/-} mice (Kayagaki et al., 2013)). Future studies should address CL actions in such a model, to definitively rule out the possibility that CL inhibits in vivo CASP11 signalling via suppressing TLR4-dependent CASP11 priming.”

-The last paragraph describing Fig. 4 is probably better suited for being part of the "Conclusions and Perspectives".

We thank the referee for this excellent suggestion, and we have moved the paragraph to “Conclusions and Perspectives”.

References

- Gandek, T. B., van der Koog, L., & Nagelkerke, A. (2023). A Comparison of Cellular Uptake Mechanisms, Delivery Efficacy, and Intracellular Fate between Liposomes and Extracellular Vesicles. *Advanced Healthcare Materials*, 12(25). <https://doi.org/10.1002/adhm.202300319>
- Gonzalvez, F., Schug, Z. T., Houtkooper, R. H., MacKenzie, E. D., Brooks, D. G., Wanders, R. J. A., Petit, P. X., Vaz, F. M., & Gottlieb, E. (2008). Cardiolipin provides an essential activating platform for caspase-8 on mitochondria. *The Journal of Cell Biology*, 183(4), 681–696. <https://doi.org/10.1083/jcb.200803129>
- Ikon, N., Su, B., Hsu, F.-F., Forte, T. M., & Ryan, R. O. (2015). Exogenous cardiolipin localizes to mitochondria and prevents TAZ knockdown-induced apoptosis in myeloid progenitor cells. *Nikita*, 464(2). <https://doi.org/10.1016/j.bbrc.2015.07.012>
- Kayagaki, N., Wong, M. T., Stowe, I. B., Ramani, S. R., Gonzalez, L. C., Akashi-Takamura, S., Miyake, K., Zhang, J., Lee, W. P., Muszynski, A., Forsberg, L. S., Carlson, R. W., & Dixit, V. M. (2013). Noncanonical inflammasome activation by intracellular LPS independent of TLR4. *Science*, 341(6151), 1246–1249. <https://doi.org/10.1126/science.1240248>
- Kjeken, R., MOUSAVI, S. A., BRECH, A., GRIFFITHS, G., & BERG, T. (2001). Wortmannin-sensitive trafficking steps in the endocytic pathway in rat liver endothelial cells. *Biochemical Journal*, 357(2), 497. <https://doi.org/10.1042/0264-6021:3570497>
- Koivusalo, M., Welch, C., Hayashi, H., Scott, C. C., Kim, M., Alexander, T., Touret, N., Hahn, K. M., & Grinstein, S. (2010). Amiloride inhibits macropinocytosis by lowering submembranous pH and preventing Rac1 and Cdc42 signaling. *The Journal of Cell Biology*, 188(4), 547–563. <https://doi.org/10.1083/jcb.200908086>
- Kunsmann, L., Rüter, C., Bauwens, A., Greune, L., Glüder, M., Kemper, B., Fruth, A., Wai, S. N., He, X., Lloubes, R., Schmidt, M. A., Dobrindt, U., Mellmann, A., Karch, H., & Bielaszewska, M. (2015). Virulence from vesicles: Novel mechanisms of host cell injury by *Escherichia coli* O104:H4 outbreak strain. *Scientific Reports*, 5(1), 13252. <https://doi.org/10.1038/srep13252>
- Li, L., Wan, T., Wan, M., Liu, B., Cheng, R., & Zhang, R. (2015). The effect of the size of fluorescent dextran on its endocytic pathway. *Cell Biology International*, 39(5), 531–539. <https://doi.org/10.1002/cbin.10424>
- Man, S. M., Tourlomousis, P., Hopkins, L., Monie, T. P., Fitzgerald, K. A., & Bryant, C. E. (2013). Salmonella Infection Induces Recruitment of Caspase-8 to the Inflammasome To Modulate IL-1 β

Production. *The Journal of Immunology*, 191(10), 5239–5246.
<https://doi.org/10.4049/jimmunol.1301581>

- Matthaeus, C., & Taraska, J. W. (2021). Energy and Dynamics of Caveolae Trafficking. *Frontiers in Cell and Developmental Biology*, 8. <https://doi.org/10.3389/fcell.2020.614472>
- Nel, A. E., Mädler, L., Velegol, D., Xia, T., Hoek, E. M. V., Somasundaran, P., Klaessig, F., Castranova, V., & Thompson, M. (2009). Understanding biophysicochemical interactions at the nano–bio interface. *Nature Materials*, 8(7), 543–557. <https://doi.org/10.1038/nmat2442>
- Newton, K., Wickliffe, K. E., Maltzman, A., Dugger, D. L., Reja, R., Zhang, Y., Roose-Girma, M., Modrusan, Z., Sagolla, M. S., Webster, J. D., & Dixit, V. M. (2019). Activity of caspase-8 determines plasticity between cell death pathways. *Nature*, 575(7784), 679–682. <https://doi.org/10.1038/s41586-019-1752-8>
- Parker, H., Chitcholtan, K., Hampton, M. B., & Keenan, J. I. (2010). Uptake of *Helicobacter pylori* Outer Membrane Vesicles by Gastric Epithelial Cells. *Infection and Immunity*, 78(12), 5054–5061. <https://doi.org/10.1128/IAI.00299-10>
- Pizzuto, M., Lonez, C., Baroja-Mazo, A., Martínez-Banaclocha, H., Tourlomousis, P., Gangloff, M., Pelegrin, P., Ruyschaert, J.-M. M., Gay, N. J., & Bryant, C. E. (2019). Saturation of acyl chains converts cardiolipin from an antagonist to an activator of Toll-like receptor-4. *Cellular and Molecular Life Sciences*, 76(18), 1–12. <https://doi.org/10.1007/s00018-019-03113-5>
- Popescu, L. M., Gherghiceanu, M., Mandache, E., & Cretoiu, D. (2006). Caveolae in smooth muscles: nanocontacts. *Journal of Cellular and Molecular Medicine*, 10(4), 960–990. <https://doi.org/10.1111/j.1582-4934.2006.tb00539.x>
- Popov, L. (2022). Mitochondrial-derived vesicles: Recent insights. *Journal of Cellular and Molecular Medicine*, 26(12), 3323–3328. <https://doi.org/10.1111/jcmm.17391>
- Rose, J. R., Christ, W. J., Bristol, J. R., Kawata, T., & Rossignol, D. P. (1995). Agonistic and antagonistic activities of bacterially derived *Rhodobacter sphaeroides* lipid A: comparison with activities of synthetic material of the proposed structure and analogs. *Infection and Immunity*, 63(3), 833–839. <https://doi.org/10.1128/iai.63.3.833-839.1995>
- Salloum, G., Bresnick, A. R., & Backer, J. M. (2023). Macropinocytosis: mechanisms and regulation. *Biochemical Journal*, 480(5), 335–362. <https://doi.org/10.1042/BCJ20210584>
- Tan, Y., Zanoni, I., Cullen, T. W., Goodman, A. L., & Kagan, J. C. (2015). Mechanisms of Toll-like Receptor 4 Endocytosis Reveal a Common Immune-Evasion Strategy Used by Pathogenic and Commensal Bacteria. *Immunity*, 43(5), 909–922. <https://doi.org/10.1016/j.immuni.2015.10.008>
- Vanaja, S. K., Russo, A. J., Behl, B., Banerjee, I., Yankova, M., Deshmukh, S. D., & Rathinam, V. A. K. (2016). Bacterial Outer Membrane Vesicles Mediate Cytosolic Localization of LPS and Caspase-11 Activation. *Cell*, 165(5), 1106–1119. <https://doi.org/10.1016/j.cell.2016.04.015>
- Vince, J. E., Nardo, D. De, Gao, W., Vince, A. J., Hall, C., McArthur, K., Simpson, D., Vijayaraj, S., Lindqvist, L. M., Bouillet, P., Rizzacasa, M. A., Man, S. M., Silke, J., Masters, S. L., Lessene, G., Huang, D. C. S., Gray, D. H. D., Kile, B. T., Shao, F., & Lawlor, K. E. (2018). The Mitochondrial Apoptotic Effectors BAX/BAK Activate Caspase-3 and -7 to Trigger NLRP3 Inflammasome and

Caspase-8 Driven IL-1 β Activation. *Cell Reports*, 25(9), 2339-2353.e4.
<https://doi.org/10.1016/j.celrep.2018.10.103>

Walsh, C., Gangloff, M., Monie, T., Smyth, T., Wei, B., McKinley, T. J., Maskell, D., Gay, N., & Bryant, C. (2008). Elucidation of the MD-2/TLR4 Interface Required for Signaling by Lipid IVa. *The Journal of Immunology*, 181(2), 1245–1254. <https://doi.org/10.4049/jimmunol.181.2.1245>

Dear Malvina,

Thank you again for submitting your revised manuscript (EMBOJ-2025-120114R) to The EMBO Journal for our consideration, and for your patience during peer review. Your manuscript has now been seen by the two original referees who had previously assessed the initial version, and I am glad to say that they both find the initially raised criticisms and concerns adequately and sufficiently addressed, and now support publication of the study in our journal without any further comments (their reports are included below).

In light of this expert input, I am very pleased to say that your manuscript has been in principle accepted for publication in The EMBO Journal. Congratulations on an excellent study!

From the editorial side, there are a few remaining changes we need you to make in a final version of the manuscript before we can proceed with publication of the article:

- Please move the Funding information to the Acknowledgements section of the manuscript.
- Please provide a list of up to 5 relevant keywords to enhance search engine discoverability of the paper after the Abstract of your revised manuscript.
- Please change the heading "Conflict of Interest" to "Disclosure and competing interests statement".
- The author contributions statement should be removed from the manuscript file. Instead, we use CRediT to specify the contributions of each author in the journal submission system. Please feel free to use the free text box to provide more detailed descriptions during submission. See also our guide to authors for more information: <https://www.embopress.org/page/journal/14602075/authorguide#authorshipguidelines>.
- The "Supplemental Information" section should be removed from the main manuscript file.
- "Declaration of generative AI and AI-assisted technologies in the writing process" should be included in the "Methods" section.
- The heading of the Appendix PDF file's title page should be "Appendix for" followed by the manuscript title and a Table of Contents including page numbers.
- Please note that EMBO press papers are accompanied online by:
 - A) a short (2 sentences) summary of the findings and their significance,
 - B) 2-5 short bullet points highlighting the key results, and
 - C) a synopsis image in .jpg or .png format that is exactly 550 pixels wide and 300-600 pixels high (the height is variable). Please note that the text needs to be legible at the final size.Please upload this information along with your revised manuscript (the text for A and B should be provided in a separate Word file).
- Please provide the exact p values in the legends of Figures 1E, H; 2G, 4E, EV1 B, EV4 B.

Please also note that as part of the EMBO publications' Transparent Editorial Process, The EMBO Journal publishes online a Peer Review File along with each accepted manuscript. This File will be published in conjunction with your paper and will include the referee reports, your point-by-point response and all pertinent correspondence relating to the manuscript. You can opt out of this by letting the editorial office know (contact@embojournal.org). If you do opt out, the Peer Review File link will point to the following statement: "No Peer Review File is available with this article, as the authors have chosen not to make the review process public in this case."

We look forward to seeing a final version of your manuscript as soon as possible. Please let us know if you have any questions and use this link to submit your revision: <https://emboj.msubmit.net/cgi-bin/main.plex>.

Best regards,

Ioannis

Ioannis Papaioannou, PhD
Editor, The EMBO Journal

i.papaioannou@embojournal.org

Referee #1:

In this revised study, the authors have addressed all my concerns. I have no additional critiques to offer.

Referee #2:

This is a highly improved version of an already originally strong paper. I appreciated new experiments performed that not only support conclusions but in some cases, such as endocytic regulation of CL uptake, further expand the impact of the findings of the authors. I also appreciate that experiments not performed led to a through discussion in the paper and to highlighting some limitations of the study. Finally, I thank the authors for sharing data on caspase-8, and I agree that, based on their preliminary findings, these are better suited for a future paper.

All editorial and formatting issues were resolved by the authors.

Dear Malvina,

Congratulations on an excellent work! I am very pleased to inform you that your manuscript has been accepted for publication in The EMBO Journal. Thank you very much for comprehensively addressing the initially raised referees' criticisms and concerns, and the editorial requests for changes and corrections.

If you have any questions, please do not hesitate to contact the Editorial Office. Thank you for your contribution to The EMBO Journal. Working with you has been a pleasure!

Best regards,

Ioannis
